# NEAR: Neural Electromagnetic Array Response

**Yinyan Bu** [1]   **Jiajie Yu** [1]   **Kai Zheng** [1]   **Xinyu Zhang** [1]   **Piya Pal** [1]

## Abstract

We address the challenge of achieving angular super-resolution in multi-antenna radar systems that are widely used for localization, navigation, and automotive perception. A multi-antenna radar achieves very high resolution by computationally creating a large virtual sensing system using very few physical antennas. However, practical constraints imposed by hardware, noise, and a limited number of antennas can impede its performance. Conventional supervised learning models that rely on extensive pre-training with large datasets, often exhibit poor generalization in unseen environments. To overcome these limitations, we propose NEAR, an untrained implicit neural representation (INR) framework that predicts radar responses at unseen locations from sparse measurements, by leveraging latent harmonic structures inherent in radar wave propagation. We establish new theoretical results linking antenna array response to expressive power of INR architectures, and develop a novel physics-informed and latent geometry-aware regularizer. Our approach integrates classical signal representation with modern implicit neural learning, enabling high-resolution radar sensing that is both interpretable and generalizable. Extensive simulations and real-world experiments using radar platforms demonstrate NEAR's effectiveness and its ability to adapt to unseen environments.

## 1. Introduction

In addition to Lidar and RGB-cameras, Radar has emerged as a crucial sensing modality for advanced sensing tasks such as driver assistance systems (ADAS) and autonomous vehicles (Bijelic et al., 2020; Caesar et al., 2020), especially

[1]Department of Electrical and Computer Engineering, University of California San Diego (UCSD), La Jolla, United States. Correspondence to: Yinyan Bu <y1bu@ucsd.edu>, Jiajie Yu <jiy088@ucsd.edu>.

*Proceedings of the 42nd International Conference on Machine Learning*, Vancouver, Canada. PMLR 267, 2025. Copyright 2025 by the author(s).

due to its robustness to adverse weather conditions (e.g. fog, snow, rain). Multiple-Input-Multiple-Output (MIMO) radar (Li & Stoica, 2008) employs an array of transmit (Tx) antennas which generate signals that are reflected by targets of interest, and received at a receiving (Rx) antenna array. The distance and velocity of these targets are characterized by using the radar's Range-Doppler (RD) map, which is computed by applying Discrete-Time Fast Fourier Transform (FFT) on digitized receiver signals after Analog-to-Digital Conversion (ADC) (Sun et al., 2020). Direction-of-Arrival (DOA) estimation is then performed exclusively on peaks that pass the constant false alarm rate (CFAR) detector (Scharf & Demeure, 1991) to determine the angular orientation of objects. The angular resolution of MIMO radar, which reveals how well it can identify two or more closely spaced sources, is fundamentally constrained by the number and configuration of the antenna array. For instance, a device equipped with eight uniformly filled antenna arrays achieves at most an angular resolution of about $15°$ (Instruments, 2017). Thus, it is important to develop innovative technologies to enhance the angular resolution of radar sensing, without incurring substantial hardware costs.

For MIMO radar, range and Doppler resolution can be improved by adjusting signal bandwidth and frame time, which correspond to the frequency range and duration of signal pulses, respectively (Li et al., 2023). However, angular resolution is strictly dependent on the radar hardware specifications, and cannot be improved through parameter adjustments. Achieving higher angular resolution in both azimuth and elevation requires a large aperture in both horizontal and vertical directions, which, for uniformly filled arrays, necessitates a significant number of antennas, resulting in high hardware costs. An efficient alternative approach, which is becoming increasingly relevant for next-generation sensing (such as automotive radars) is to use sparse arrays (Pal & Vaidyanathan, 2010; Qin et al., 2015; Sarangi et al., 2023). Sparse arrays deploy a reduced number of transmit and receive antennas in order to achieve the same aperture as a standard uniform array (with quadratically larger number of sensors), which necessitates larger inter-element spacings. They are designed so that their virtual (sum or difference) co-arrays are dense uniform arrays filling the available aperture. This property can be utilized in several ways, such as localization of more sources than sensors, and achieving

very high resolution with sufficient temporal measurements (Cheng et al., 2014; Liu & Vaidyanathan, 2017; Wang & Nehorai, 2017; Qiao & Pal, 2019). Recently, it has been shown that sparse arrays are also near-optimal subspace codes, highlighting their novel connection to channel coding (Mahdavifar et al., 2024). However, naive processing of sparse array outputs using traditional or ad-hoc methods, can suffer from high sidelobes and degrade DOA estimation accuracy (Sun & Zhang, 2021). Thus, achieving super-resolution angular resolution with low hardware cost and irregular sampling geometries, remains a continuing challenge.

In this work, we introduce a machine learning framework that tackles the challenge of angular super-resolution at low hardware cost using sparse measurements that employ only a few antennas. Our goal is to predict complex-valued responses at any desired location (that can potentially be used for DOA estimation) within the 2D *virtual* antenna array domain using only a sparse set of responses. One straightforward approach to accomplish this is to train a machine learning model that maps a spatial location to the corresponding antenna response. However, this approach may fail to incorporate the underlying physics of wave signal propagation, and thus still require very dense measurements to achieve reasonable performance. As one of the important breakthroughs in computer vision, NeRF (Mildenhall et al., 2021) has achieved remarkable success in 3D reconstruction and view synthesis tasks by learning a scene's radiance field from a set of input images and generating photorealistic renderings from novel viewpoints. At its core, NeRF utilizes implicit neural representations (INRs) (Sitzmann et al., 2020; Tancik et al., 2020) to parameterize the radiance field as a continuous function, modeled by a multilayer perceptron (MLP) that maps 3D spatial coordinates to RGB color and volume density. Leveraging volume-rendering techniques to synthesize images, NeRF incorporates the underlying physics of light propagation.

Inspired by recent advances in NeRF and INRs, we propose Neural Electromagnetic Array Response (NEAR), a framework utilizing INR, that maps 2D spatial coordinates to complex-valued antenna response at those locations. Several distinguishing features differentiate our task from traditional NeRF applications, particularly due to fundamental differences between the ways in which visible light and radar signals are processed (Zhao et al., 2023). Firstly, we have access to a limited number of complex-valued measurements, proportional to the number of deployed antennas. Apparently, this conveys significantly less information compared to an image comprising thousands of pixels. As a consequence, training a model using off-the-shelf INR-based algorithms with a limited number of antennas can fail to reliably predict unseen array response at arbitrary spatial locations. Secondly, while optical NeRF frameworks rely solely on light intensity (amplitude), radar signals at

millimeter wavelengths necessitate consideration of phase information. Unlike visible light, where the phase is often neglected, the phase in radar signals is crucial for capturing fine-grained details of wave propagation, such as target locations. Ignoring phase would result in a significant loss of critical information. Finally, despite the increasing adoption of INRs in various domains, the theoretical understanding of their properties and their implications in specific applications remain limited. Key aspects, such as the behavior of deep layers in these networks and role of positional encoding (PE) in representing complex signals, are not yet well-understood.

To address these challenges, our work makes the following contributions:

- We propose NEAR, the *first* electromagnetic array response prediction framework that implicitly integrates signal propagation characteristics into INRs. Our approach enables prediction of array response at unseen receiver locations, facilitating super-resolution angular estimation with low hardware requirements.

- We provide tight characterization of the class of functions that INR's can represent with certain choices of positional encoding and activation functions. Our results improve upon existing theoretical analysis of INRs.

- We evaluate NEAR through both simulation studies and real-world experiments, achieving superior performance in antenna array response prediction and other downstream tasks such as super-resolution angular estimation compared to existing model-based and machine learning methods.

Overall, we believe our findings contribute to advancing research in INRs and their unique applications in radar sensing. Our work also marks the first step towards leveraging INRs for predicting unseen antenna responses in radar sensing, paving the way for new opportunities to enhance the performance of future sensing and localization systems.

## 2. Preliminaries

In this section, we provide background knowledge on MIMO radar, virtual array, and implicit neural representation.

**MIMO Radar.** We consider targets in three-dimensional Euclidean space, represented by spherical coordinates as depicted in Figure 1. We consider a L-shaped MIMO radar system (consistent with our hardware) with $N_t$ physical Tx antennas located at $\{(x_{T,1}, 0, 0), \cdots, (x_{T,N_t}, 0, 0)\}$ and $N_r$ physical Rx antennas located at $\{(0, y_{R,1}, 0), \cdots, (0, y_{R,N_r}, 0)\}$. The

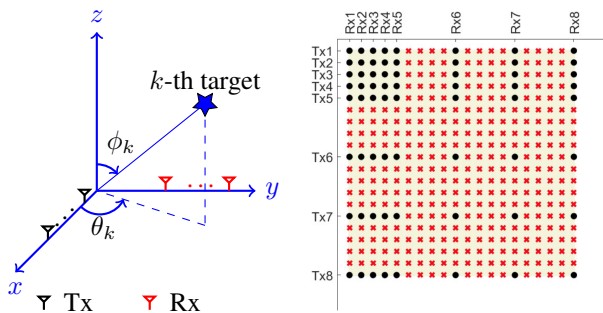

*Figure 1.* Left: Illustration of the target position in Spherical coordinate system. Right: One sub-Nyquist sampling pattern with ✗ indicating missing virtual element response.

Tx antennas emit a set of $N_t$ orthogonal waveforms, which are reflected by $K$ targets and their superposition is received at the Rx antenna array. Each Rx antenna of a MIMO radar is equipped with a bank of $N_t$ matched filters, each matched to one of the $N_t$ orthogonal waveforms. This yields a total of $N_t N_r$ measurements at the output of $N_t N_r$ matched filters, which can be used to perform different spatial sensing tasks such as localization, beamforming and so forth (Li & Stoica, 2008).

***Virtual* Array.** One of the key features of a MIMO radar is that by using only $N_t + N_r$ physical Tx and Rx antennas, it can create the effect of a much larger antenna array with $N_t N_r$ *virtual* sensing elements at the output of the $N_t N_r$ matched filters. Consider a far-field point target at direction $(\theta, \phi)$. It can be shown that the noiseless array response at the $m$-th matched filter output in the $n$-th receiving antenna can be expressed as

$$x e^{j \frac{2\pi}{\lambda} (x_{T,m} \sin \phi_k \cos \theta_k + y_{R,n} \sin \phi_k \sin \theta_k)}, \quad (1)$$

where $x$ is the unknown amplitude of the signal reflected by the target and $\lambda$ is the wavelength at which the narrowband radar operates. Therefore, the array response in (1) is the same as that of a (fictitious) two-dimensional receiving array with $N_t N_r$ antenna elements located at

$$\{(x_{T,m}, 0, 0) + (0, x_{R,n}, 0), 1 \le m \le N_t, 1 \le n \le N_r\}.$$

This two-dimensional antenna array with $N_t N_m$ elements is known as the *virtual array* (Chen & Vaidyanathan, 2008). Figure 1 shows a physical Tx-Rx antenna pair and the associated two-dimensional virtual array. Notice that depending on the geometry of the Tx-Rx pair, the 2D virtual array need not comprise of elements on consecutive locations over a uniform grid, and there can be missing elements (or holes) the virtual array, as indicated in Figure 1.

Consider $K$ targets in the far field, with azimuth angle $\theta_k$ and elevation angle $\phi_k, 1 \le k \le K$. Without loss of generality, let the reference *virtual* antenna be located at the

origin of the coordinate system. The array response at a coordinate $(r_1, r_2)$ [1] (which may be a virtual array element location, or the location of a missing sensor), due to signals impinging from the $K$ targets in absence of noise can be expressed as:

$$y_{r_1, r_2} = \sum_{k=1}^{K} x_k e^{j \frac{2\pi}{\lambda} (r_1 \sin \phi_k \cos \theta_k + r_2 \sin \phi_k \sin \theta_k)}, \quad (2)$$

where $x_k$ is the unknown complex-valued reflection coefficient of $k$-th target. Various algorithms, such as beamforming (Van Trees, 2002) and subspace-based methods (Schmidt, 1986; Roy & Kailath, 1989), can be applied to estimate $\{\theta_k\}_{k=1}^{K}$ and $\{\phi_k\}_{k=1}^{K}$.

**Implicit Neural Representations (INR).** INRs are used to model a continuous function $g : \mathbb{R}^{d_{in}} \to \mathbb{R}^{d_{out}}$ using a neural network $f_{\Theta} : \mathbb{R}^{d_{in}} \to \mathbb{R}^{d_{out}}$, parameterized by weights $\Theta$, which map input coordinates $\boldsymbol{r} \in \mathbb{R}^{d_{in}}$ to signal values $g(\boldsymbol{r}) \in \mathbb{R}^{d_{out}}$. A significant challenge for INRs is to accurately reconstruct high-frequency details, which is needed for radar super-resolution. Classical neural network architectures are known to exhibit strong spectral bias (Rahaman et al., 2019) towards lower frequencies. Recently, Tancik et al. (2020); Sitzmann et al. (2020) have proposed architectural solutions to overcome this spectral bias allowing faster convergence and higher accuracy of INRs.

Following the formulation of (Yüce et al., 2022), most INR architectures can be decomposed into a mapping function $\gamma : \mathbb{R}^D \to \mathbb{R}^T$ followed by a MLP, with weights $\boldsymbol{W}^{(\ell)} \in \mathbb{R}^{F_\ell \times F_{\ell-1}}$, bias $\boldsymbol{b}^{(\ell)} \in \mathbb{R}^{F_\ell}$, and activation function $\rho^{(\ell)} : \mathbb{R} \to \mathbb{R}$ applied element-wise at each layer $\ell = 1, \ldots, L-1$. Suppose $\boldsymbol{z}^{(\ell)}$ represents the post activation output at layer $\ell$. The INR input-output relationship is given by

$$\boldsymbol{z}^{(0)} = \gamma(\boldsymbol{r}),$$
$$\boldsymbol{z}^{(\ell)} = \rho^{(\ell)} \left( \boldsymbol{W}^{(\ell)} \boldsymbol{z}^{(\ell-1)} + \boldsymbol{b}^{(\ell)} \right), \quad \ell = 1, \ldots, L-1,$$
$$f_{\Theta}(\boldsymbol{r}) = \boldsymbol{W}^{(L)} \boldsymbol{z}^{(L-1)} + \boldsymbol{b}^{(L)}. \quad (3)$$

Tancik et al. (2020) introduced Fourier feature networks (FFNs), which use Fourier-based positional encoding $\gamma(\boldsymbol{r}) = \sin(\boldsymbol{\Omega}\boldsymbol{r} + \boldsymbol{\phi})$, with parameters $\boldsymbol{\Omega} \in \mathbb{R}^{T \times D}$ and $\boldsymbol{\phi} \in \mathbb{R}^T$ followed by an MLP with $\rho^{(\ell)} = \text{ReLU}$. They demonstrated that by initializing $\boldsymbol{\Omega}_{i,j} \sim \mathcal{N}(0, \sigma^2)$ with random Fourier features, and choosing large values of $\sigma$, one can drive the network response towards realizing higher frequencies. SIREN (Sitzmann et al., 2020) can also mitigate spectral bias in a similar way by choosing a different (sinusoidal) activation function and rescaling certain parameters at initialization.

---

[1] In our setting, the 2D virtual array is located in the $xy$-plane.

## 3. Related Work

**Hallucinated Antenna Interpolation/Extrapolation.** To mitigate the high sidelobes introduced by the sparse arrays and enhance the SNR of antenna array response, Sun & Zhang (2021) propose to recover missing elements or holes in the sparse arrays by completing a low-rank (Block) Hankel matrix (Chen & Chi, 2013). However, the nuclear norm minimization that they employed often suffers from suboptimal recovery performance (Lu et al., 2015) and exhibits sensitivity to the sampling pattern and noise(Bu et al., 2025; Sarangi et al., 2022). Furthermore, their approach is limited in its applicability to cases involving non-integer-multiple sampling. To enhance the azimuth resolution of MIMO radar, Li et al. (2023) proposed Analog-to-Digital super-resolution model (ADC-SR) that predicts or hallucinates additional radar signals using signals from only a few receivers, essentially implementing a uniformly filled array extrapolation framework. However, their approach is restricted to 1D MIMO configurations and relies on a large training dataset, potentially limiting its generalization capability. In contrast, our method implicitly leverages the underlying physics of signal propagation and requires only single-snapshot sparse measurements, eliminating the dependence on extensive training data.

**Expressive power of INRs.** INRs have emerged as a versatile set of neural architectures for representing and processing signals on low-dimensional spaces. Understanding the function class that an INR architecture can represent is essential for their application to practical problems. Recognizing that polynomials of sinusoids generate linear combinations of integer harmonics of said sinusoids, Yüce et al. (2022) analyzed the expressivity of FFN, SIREN and related architectures in (Fathony et al., 2020). Subsequently Roddenberry et al. (2023) developed a broader theoretical understanding of INR architectures with a wider class of activation functions and provided a *superset* to which the integer harmonic frequencies characterizing INR functions belong. While their *superset* results provide valuable theoretical insights, our work refines this analysis and derives the exact set (and not a superset) of integer harmonics which describe the expressive power of INRs, delivering a tight characterization.

**Neural Radio-Frequency Field Reconstruction.** Building on the fact that light is a kind of electro magnetic (EM) wave, Zhao et al. (2023) and (Lu et al.) proposed two NeRF-based frameworks, named NeRF[2] and NeWRF, respectively, for wireless channel modeling based on implicit wireless radiation field reconstruction. Chen et al. (2024) further developed a hybrid model that integrates NeRF-like object representation with physics-based ray tracing models. These models enable accurate characterization and prediction of channel properties. Building on the principles of planar wave propagation, we propose a novel framework for reconstructing 2D MIMO *virtual* antenna array response fields using implicit neural representations. In contrast to aforementioned approaches that employ ray tracing for EM waves, our method employs a straightforward yet effective regularization strategy specifically designed to leverage the spectral sparsity of antenna array measurements and the characteristics of planar wave propagation.

## 4. Neural Electromagnetic Array Response

In this section, we present the design of NEAR. Section 4.1 outlines our problem formulation, followed by theoretical results on the expressive power of INRs and their connection to Fourier series in Section 4.2 elucidating why and how array response function in (2) can be effectively learned. Section 4.3 details our novel implicit regularization strategy that integrates signal propagation model while harnessing harmonic structure. Finally, we describe the response prediction process of NEAR in Section 4.4.

### 4.1. Problem Formulation

We consider a environment where all objects are located in the far-field relative to the radar antenna array. In this setup, the propagation of wireless signals can be modeled as planar waves that are emitted from the Tx array, reflected by objects and finally captured by the Rx array. Let $\mathcal{S}_x$ and $\mathcal{S}_y$ represent the sparse sets of physical Tx and Rx antennas, respectively. The coordinate set of available *virtual* antennas is given by $\{(r_x, r_y)\}_{r_x \in \mathcal{S}_x, r_y \in \mathcal{S}_y}$ as explained in Section 2. We define the domain of antenna array response field as $\mathcal{D} = \{(x, y) \mid 0 \leq x \leq \max(\mathcal{S}_x), 0 \leq y \leq \max(\mathcal{S}_y)\}$. We represent the continuous complex-valued response field as a function $y : \mathcal{D} \to \mathbb{C}$, where the input is a 2D coordinate $\boldsymbol{r} = [r_1, r_2]^\top$ within the domain $\mathcal{D}$, and the output is a complex-valued response $y_{r_1, r_2}$ that adheres to the signal model in (2). To approximate this continuous 2D response field, we employ an INR model that maps the input 2D coordinates to a vector in $\mathbb{R}^2$, where the two components correspond to the real and imaginary parts of the complex-valued response, respectively. Specifically, the model is defined as $f_\Theta : \mathbb{R}^2 \to \mathbb{R}^2$, and the parameters $\Theta$ are optimized to map each input 2D coordinate to its corresponding response. *The goal of this paper is to learn the function $f_\Theta$ solely from physical antenna measurements (without using any offline training data), by exploiting the harmonic structure of array response in (2).* Once the response function is learned, it enables the prediction of array responses at any unseen locations within $\mathcal{D}$, facilitating downstream tasks such as angle estimation and localization.

## 4.2. Representational Ability of INRs

While substantial empirical evidence demonstrates the effectiveness of INRs in representing scenes and various visual signals, the theoretical underpinnings of their ability to approximate continuous functions remain underexplored. In this subsection, we establish that many contemporary INRs inherently build upon similar underlying structures and shared fundamental principles, enabling them to represent a certain class of signals.

To rigorously analyze the expressive power of INRs, we follow the formulation outlined in (3). Following (Yüce et al., 2022; Roddenberry et al., 2023; Mehmeti-Göpel et al., 2020), we restrict our investigation to polynomial activation functions of the form $\rho(x) = \sum_{q=0}^{Q} \alpha_q x^q$, a widely adopted approach in the study of the expressive capacity of INRs.

**Theorem 4.1.** *Let $f_{\Theta} : \mathbb{R}^D \to \mathbb{R}$ be an INR given by* (3), *where the activation function for layers $\ell > 1$ is given by $\rho^{(\ell)}(z) = \sum_{q=0}^{Q} \alpha_q z^q$. Let $\Omega_{\mathcal{T}} = [\boldsymbol{\omega}_1, \ldots, \boldsymbol{\omega}_T]^{\top} \in \mathbb{R}^{T \times D}$ represent the frequency matrix and $\boldsymbol{\phi}_{\mathcal{T}} \in \mathbb{R}^T$ the phase vector used to map the input coordinate $\boldsymbol{r} \in \mathbb{R}^D$ into the feature space via the mapping $\gamma(\boldsymbol{r}) = \sin(\Omega_{\mathcal{T}} \boldsymbol{r} + \boldsymbol{\phi}_{\mathcal{T}})$. The resulting architecture is capable of representing functions of the form:*

$$f_{\Theta}(\boldsymbol{r}) = \sum_{\boldsymbol{s} \in \mathcal{S}_{\mathcal{T}}} c_{\boldsymbol{s}} \sin\left(\langle \Omega_{\mathcal{T}}^{\top} \boldsymbol{s}, \boldsymbol{r} \rangle + \phi_{\boldsymbol{s}}\right), \qquad (4)$$

*where*

$$\mathcal{S}_{\mathcal{T}} = \left\{ [s_1, s_2, \ldots, s_T]^{\top} \;\middle|\; s_t \in \mathbb{Z}, \; \sum_{t=1}^{T} |s_t| \leq Q^{L-1} \right\}.$$

Theorem 4.1 gives an *exact* characterization of the set $\mathcal{S}_{\mathcal{T}}$ of all possible integer harmonics of the feature mapping $\gamma(\boldsymbol{r})$. In contrast, Yüce et al. (2022); Roddenberry et al. (2023) only provide a superset to which $\mathcal{S}_{\mathcal{T}}$ belongs.

*Remark* 4.2. Let $y_{\text{R}}$ and $y_{\text{I}}$ denote the real and imaginary parts of the response field function (2), respectively. The function $y_{\text{R}}(\boldsymbol{r})$ can be equivalently represented as (4) by applying Theorem 4.1 with the following parameterization:

$$\Omega_{\mathcal{T}} = \frac{2\pi}{\lambda} [\sin\phi_k \cos\theta_k \;\; \sin\phi_k \sin\theta_k]_{1 \leq k \leq K} \in \mathbb{R}^{K \times 2},$$

$$c_{2k-1} = \text{Re}(x_k), \; c_{2k} = -\text{Im}(x_k), \; \phi_{2k-1} = \frac{\pi}{2},$$

$$\phi_{2k} = 0, \; \boldsymbol{s}_k = \boldsymbol{e}_k \in \mathbb{R}^K, \; \mathcal{S}_{\mathcal{T}} = \{\boldsymbol{e}_k\}_{1 \leq k \leq K}.$$

Under this parameterization,

$$y_{\text{R}}(\boldsymbol{r}) = \sum_{k=1}^{K} c_{2k-1} \sin\left(\langle \Omega_{\mathcal{T}}^{\top} \boldsymbol{s}_k, \boldsymbol{r} \rangle + \phi_{2k-1}\right)$$
$$+ c_{2k} \sin\left(\langle \Omega_{\mathcal{T}}^{\top} \boldsymbol{s}_k, \boldsymbol{r} \rangle + \phi_{2k}\right).$$

A similar representation also holds for $y_{\text{I}}$.

This shows that our desired array response indeed belongs to the class of functions representable by INRs. Although the resulting INR architecture appears deceptively simple, it is to be noted that the positional encoding requires the *ground-truth DOA and amplitude of each target*, which are never available in practice. Hence it is important to investigate the class of functions that INR can approximate using a given mapping $\gamma(\boldsymbol{r})$, such as the type of fixed sinusoid positional encodings employed in NeRF (Mildenhall et al., 2021):

$$\gamma(\boldsymbol{r}) = \begin{bmatrix} \sin(\Omega \boldsymbol{r}) \\ \cos(\Omega \boldsymbol{r}) \end{bmatrix}, \text{ with} \qquad (5)$$

$$\Omega = \begin{bmatrix} 2^0\pi & 0 & 2^1\pi & 0 & \cdots & 2^{T-1}\pi & 0 \\ 0 & 2^0\pi & 0 & 2^1\pi & \cdots & 0 & 2^{T-1}\pi \end{bmatrix}^{\top}.$$

Consider a non-periodic function $g : \mathbb{R}^{d_{in}} \to \mathbb{R}^{d_{out}}$ defined over a bounded domain $\mathcal{D}$ (e.g. the height and width of a image, the aperture of 2D-MIMO array). We can define its periodic extension $\tilde{g} : \mathbb{R}^{d_{in}} \to \mathbb{R}^{d_{out}}$ with period $\boldsymbol{p}$ as follows (Benbarka et al., 2022):

$$\tilde{g}(\boldsymbol{x} + \boldsymbol{n} \circ \boldsymbol{p}) = g(\boldsymbol{x}) \quad \forall \boldsymbol{x} \in \mathcal{D}, \; \forall \boldsymbol{n} \in \mathbb{Z}^{d_{in}}, \qquad (6)$$

where $\circ$ denotes the Hadamard product. By normalizing the input domain to its respective bounds, we assume a period of 2 for each variable, i.e., within the range $[-1, 1)$. The Fourier series expansion for a periodic extension $\tilde{g} : \mathbb{R}^2 \to \mathbb{R}$ of period $\boldsymbol{2}$ is given by (Oppenheim et al., 2010):

$$\sum_{m,n=-\infty}^{\infty} A_{m,n} \cos(\pi(mx+ny)) + B_{m,n} \sin(\pi(mx+ny)).$$
$$(7)$$

It can be shown that if the frequency matrix $\Omega \in \mathbb{R}^{2T \times 2}$ of the INR described in Theorem 4.1 is chosen according to (5), then as the number ($L$) of layers of the MLP/INR increases, $f_{\Theta}$ approximates to certain period-$\boldsymbol{2}$ functions $\tilde{g}$ of the form (7). See Appendix A.7 for more details.

## 4.3. Physics-Informed Implicit Regularization

To model the antenna array response field, we discretize the domain of interest into a finite set of points in the 2D plane. Let $[0, U_1] \times [0, U_2]$ represent the antenna array response field with bounded domain positioned in the $x - y$ plane, consider a general case of a uniform sampling grid of dimensions $M_1 \times M_2$, with spacing $d_1 = \frac{U_1}{M_1 - 1} \leq \frac{\lambda}{2}$ and $d_2 = \frac{U_2}{M_2 - 1} \leq \frac{\lambda}{2}$. Supposing an array snapshot containing $K$ targets with azimuth angle $\theta_k$ and elevation angle $\phi_k$ ($k = 1, \cdots, K$), and leveraging planar wave propagation (2), the $(m_1, m_2)$ th element of the response with respect to

$K$ targets in the absence of noise can be written as

$$
y_{m_1, m_2}
$$
$$
= \sum_{k=1}^{K} x_k e^{j\frac{2\pi}{\lambda}((m_1-1)d_1 \sin\phi_k \cos\theta_k + (m_2-1)d_2 \sin\phi_k \sin\theta_k)}
$$

$$(8)$$

for $1 \leq m_1 \leq M_1$ and $1 \leq m_2 \leq M_2$. Notably, when $d_1 = d_2 = \frac{\lambda}{2}$, the sampling pattern aligns with the Nyquist sampling. Let $\mathbf{Y} = [y_{m_1,m_2}]_{1 \leq m_1 \leq M_1, 1 \leq m_2 \leq M_2} \in \mathbb{C}^{M_1 \times M_2}$ be the ground truth response matrix with entries as the antenna array response defined in (8).

**Definition 4.3.** Given $\mathbf{Y} = [y_{m_1,m_2}] \in \mathbb{C}^{M_1 \times M_2}$ for $1 \leq m_1 \leq M_1, 1 \leq m_2 \leq M_2$, a **Block Hankel matrix** of $\mathbf{Y}, 1 \leq N_1 \leq M_1, 1 \leq N_2 \leq M_2$ can be constructed as:

$$
\mathcal{H}_{N_1,N_2}(\mathbf{Y})
$$
$$
= \begin{bmatrix}
\mathcal{H}_{N_2}(\mathbf{y}_1) & \mathcal{H}_{N_2}(\mathbf{y}_2) & \cdots & \mathcal{H}_{N_2}(\mathbf{y}_{M_1-N_1+1}) \\
\mathcal{H}_{N_2}(\mathbf{y}_2) & \mathcal{H}_{N_2}(\mathbf{y}_3) & \cdots & \mathcal{H}_{N_2}(\mathbf{y}_{M_1-N_1+2}) \\
\vdots & \vdots & \ddots & \vdots \\
\mathcal{H}_{N_2}(\mathbf{y}_{N_1}) & \mathcal{H}_{N_2}(\mathbf{y}_{N_1+1}) & \cdots & \mathcal{H}_{N_2}(\mathbf{y}_{M_1})
\end{bmatrix},
$$

where $\mathcal{H}_{N_2}(\mathbf{y}_m), 1 \leq m \leq M_1$ is defined as:

$$
\mathcal{H}_{N_2}(\mathbf{y}_m) = \begin{bmatrix}
y_{m,1} & y_{m,2} & \cdots & y_{m,M_2-N_2+1} \\
y_{m,2} & y_{m,3} & \cdots & y_{m,M_2-N_2+2} \\
\vdots & \vdots & \ddots & \vdots \\
y_{m,N_2} & y_{m,N_2+1} & \cdots & y_{m,M_2}
\end{bmatrix}.
$$

*Remark* 4.4. Definition 4.3 defines the block Hankel matrix constructed along the row direction. Similarly, a block Hankel matrix can also be constructed along the column direction, denoted as $\tilde{\mathcal{H}}_{\tilde{N}_1,\tilde{N}_2}(\mathbf{Y})$ (see definition in Appendix B.1). Moreover, the rank property remains consistent for block Hankel matrices constructed along both the row and column directions.

We emphasize that the number of targets in the same range-Doppler bin that need angle estimation is small since the targets are first separated in range-Doppler domain (Sun et al., 2020). In other words, the targets are sparsely present in the angular domain and, as a result, $\mathcal{H}_{N_1,N_2}(\mathbf{Y})$ and $\tilde{\mathcal{H}}_{\tilde{N}_1,\tilde{N}_2}(\mathbf{Y})$ exhibit low rank, with rank equal to $K$ for appropriate choice of $N_1, N_2, \tilde{N}_1, \tilde{N}_2$ (see Lemma B.1 in Appendix). To characterize such a property, numerous convex/non-convex rank surrogate functions have been explored in the literature, which include but are not limited to nuclear norm (Candes & Recht, 2012), schatten-p norm (Mohan & Fazel, 2012) and truncated nuclear norm (Hu et al., 2012). However, all of these surrogate functions are explicit and requires singular value decomposition (SVD), which can be not only computational expensive but also sub-optimal. In this work, we propose a novel implicit regularizer that exploits the structure of the block Hankel matrix

and its latent representation. To further justify the effectiveness, we establish the algebraic properties of the block Hankel matrix corresponding to the ground truth response $\mathbf{Y}$ using its harmonic structure.

**Theorem 4.5.** *Consider the ground truth response matrix $\mathbf{Y}$ as defined in* (8). *For $K \leq \min(\lceil \frac{M_1}{2} \rceil, \lceil \frac{M_2}{2} \rceil)$, there exists vectors $\boldsymbol{m}_1^o \in \mathbb{C}^K$ and $\boldsymbol{m}_2^o \in \mathbb{C}^K$ such that the last column of $\mathcal{H}_{M_1,M_2-K}(\mathbf{Y})$ can be uniquely represented in terms of the first $K$ columns of $\mathcal{H}_{M_1,M_2-K}(\mathbf{Y})$ using the corresponding coefficient vectors $\boldsymbol{m}_1^o$, i.e.*

$$
\|\mathcal{H}_{M_1,M_2-K}(\mathbf{Y})\boldsymbol{S}\mathbf{m}_1^o - \mathcal{H}_{M_1,M_2-K}(\mathbf{Y})\boldsymbol{b}\|_2 = 0,
$$

*where $\boldsymbol{S} = \begin{bmatrix} \boldsymbol{I}_{K \times K} & \boldsymbol{0}_K \end{bmatrix}^{\top} \in \mathbb{R}^{(K+1) \times K}$, and $\boldsymbol{b} = \begin{bmatrix} \boldsymbol{0}_K^{\top} & 1 \end{bmatrix}^{\top} \in \mathbb{R}^{(K+1) \times 1}$. Similarly, an equivalent property holds for $\tilde{\mathcal{H}}_{M_2,M_1-K}(\mathbf{Y})$, given by*

$$
\|\tilde{\mathcal{H}}_{M_2,M_1-K}(\mathbf{Y})\boldsymbol{S}\mathbf{m}_2^o - \tilde{\mathcal{H}}_{M_2,M_1-K}(\mathbf{Y})\boldsymbol{b}\|_2 = 0.
$$

Building upon the planar wave signal propagation model, Theorem 4.5 establishes a connection between rank property and least squares by leveraging harmonic structure of Block Hankel matrix. However, the global optimizer $\mathbf{m}_1^o$ and $\mathbf{m}_2^o$ are intrinsically dependent on parameters $\{(\theta_k, \phi_k)\}_{k=1}^{K}$ (see Lemma B.3 in Appendix), which are part of the radar sensing task and not known in advance. To address this, as detailed in the next subsection, we integrate the least squares term into the loss function and parameterize the unknown coefficients, enabling them to be learned adaptively.

### 4.4. Optimizing NEAR

In practice, the model predicts the real and imaginary parts of the response signal $(\Re\{y_{r_1,r_2}\}, \Im\{y_{r_1,r_2}\})$, instead of amplitude and phase $(A(y_{r_1,r_2}), \psi(y_{r_1,r_2}))$. This is because phase is modulo against $2\pi$, which is not differentiable. We perform uniform inference for $f_{\boldsymbol{\Theta}}(\cdot)$ over the bounded domain, using a pre-chosen grid of $M_1 \times M_2$ data points. Denote the predicted response at the $(m_1, m_2)$ th element as $\hat{y}_{m_1,m_2} = f_{\boldsymbol{\Theta}}((m_1-1)\frac{U_1}{M_1-1}, (m_2-1)\frac{U_2}{M_2-1})$, and let $\hat{\mathbf{Y}}$ represent the predicted response matrix. All the other notations remain consistent with those introduced in Section 4.3, with an additional ^ to distinguish predicted quantities. Consider two sparse sampling pattern $\mathcal{S}_x, \mathcal{S}_y$, where the observed noisy response $\tilde{y}_{r_x,r_y}$ is only available at locations $\boldsymbol{r} = [r_x, r_y] \, \forall r_x \in \mathcal{S}_x, \forall r_y \in \mathcal{S}_y$. The overall loss function is defined as

$$
\mathcal{L}(\boldsymbol{\Theta}, \mathbf{m}_1, \mathbf{m}_2) = \mathcal{L}_d + \lambda \mathcal{L}_r, \tag{9}
$$

with

$$\mathcal{L}_d = \sum_{j \in \mathcal{S}_y} \sum_{i \in \mathcal{S}_x} \|f_{\boldsymbol{\Theta}}(i,j) - \tilde{y}_{i,j}\|_2,$$

$$\mathcal{L}_r = \|(\mathcal{H}_{M_1, M_2 - K}(\hat{\mathbf{Y}})\mathbf{S}\mathbf{m}_1 - \mathcal{H}_{M_1, M_2 - K}(\hat{\mathbf{Y}})\mathbf{b}\|_2$$
$$+ \|(\tilde{\mathcal{H}}_{M_2, M_1 - K}(\hat{\mathbf{Y}})\mathbf{S}\mathbf{m}_2 - \tilde{\mathcal{H}}_{M_2, M_1 - K}(\hat{\mathbf{Y}})\mathbf{b}\|_2,$$

$$\hat{\mathbf{Y}} = [f_{\boldsymbol{\Theta}}(i,j)]_{1 \le i \le M_1, 1 \le j \le M_2}.$$

$$(10)$$

Specifically, $\mathcal{L}_d$ represents data fitting term, which quantifies the gap between the predicted and acquired responses at observed locations; $\mathcal{L}_r$ corresponds to regularization term, as elaborated in Section 4.3 and Appendix B. Parameters are optimized by minimizing the total loss function

$$\boldsymbol{\Theta}^o, \mathbf{m}_1^o, \mathbf{m}_2^o = \arg \min_{\boldsymbol{\Theta}, \mathbf{m}_1, \mathbf{m}_2} \mathcal{L}_d + \lambda \mathcal{L}_r. \quad (11)$$

Using the optimal parameters $\boldsymbol{\Theta}^o$, the predicted array response can be computed by $\hat{y}_{i,j} = f_{\boldsymbol{\Theta}^o}(i,j), \ \forall i,j \in \mathcal{D}$.

## 5. Experiments

We evaluate the performance of NEAR on both simulated (Section 5.1) and real-world (Section 5.2) tasks. All experiments are run on a laptop with CPU AMD Ryzen 9 5900 HS with Radeon Graphics and GPU NVIDIA GeForce RTX 3050 Ti Laptop. See Appendix C for more experimental results. The codes are available at: https://github.com/J1mmyYu1/NEAR.

**Baselines and Benchmark.** We compare NEAR against four representative baselines: Enhanced Matrix Completion (EMaC) (Chen & Chi, 2013), SIREN (Sitzmann et al., 2020), NeRF$^2$ (Zhao et al., 2023), and NEAR without Regularization (NEAR w/o R), more implementation details and analysis of these baseline methods can be found in Appendix C.2. For a fair comparison, we adopt the hyperparameters recommended by the original authors. Additionally, we include a $20 \times 20$ full *virtual* array response (noisy) as a benchmark reference.

**NEAR Architecture.** In both simulated and real-world settings, we employ the architecture described in Equation (3), with a depth of $L = 4$, ReLU activation function $\rho(\cdot) = \text{ReLU}(\cdot)$, and positional encoding $\gamma(\boldsymbol{r})$ following NeRF's formulation in Equation (5) with $T = 10$. The hidden layer dimension is set to 256. Additional implementation details and hyperparameter configurations are provided in Appendix C.1.

### 5.1. Simulation Tasks

**Response Recovery.** We evaluate the response recovery performance of NEAR against baseline methods and the full *virtual* array benchmark, as summarized in Tables 1 - 3. The evaluation metric is the Normalized Root Mean

Square Error (NRMSE), defined as $\frac{1}{N} \sum_{n=1}^{N} \frac{\|\hat{Y}_n - Y_n\|_F}{\|Y_n\|_F}$, where $\hat{Y}_n$ and $Y_n$ denote the predicted array response and the (noiseless) ground truth full *virtual* array response at $n$-th realization, respectively, with $\| \cdot \|_F$ representing the Frobenius norm. Our method consistently outperforms all baselines across different evaluation settings, demonstrating superior generalization in response recovery tasks. Notably, NEAR achieves even lower error than the $20 \times 20$ full *virtual* array benchmark across different SNR levels with a fixed sampling number (Table 1). This can be attributed to the inherent denoising ability of our regularizer that exploits low-dimensional structure of array response and provides a cleaner estimate of at a given coordinate, compared to actual noisy measurement at the same location. The poor performance of SIREN and NEAR w/o R across all settings suggests that these models struggle to learn the appropriate continuous response function in the absence of physics-informed regularization. This highlights the importance of incorporating prior knowledge into implicit neural representations for structured signal recovery. A more detailed analysis is provided in the Ablation Study.

*Table 1.* Averaged NRMSE of response at different SNR level. 8×8 sampling is employed for NEAR, EMaC, NEAR w/O R and SIREN.

| METHOD | 10 dB | 20 dB | 30 dB |
|---|---|---|---|
| BENCHMARK | 0.2608 | 0.0825 | 0.0261 |
| NEAR | **0.2248** | **0.0495** | **0.0189** |
| EMaC | 0.3537 | 0.1889 | 0.0921 |
| NEAR w/O R | 1.0663 | 1.0504 | 1.0485 |
| SIREN | 1.0512 | 1.0277 | 1.0244 |

*Table 2.* Averaged NRMSE of response for different sampling number at 20 dB with 2 targets.

| METHOD | 6x6 | 8x8 | 10x10 |
|---|---|---|---|
| NEAR | **0.1884** | **0.0495** | **0.0362** |
| EMAC | 0.5306 | 0.1889 | 0.0724 |
| NEAR w/O R | 1.0689 | 1.0504 | 1.0030 |
| SIREN | 1.0656 | 1.0277 | 0.9860 |

*Table 3.* Averaged NRMSE of response for different number of targets at 20 dB. 8×8 sampling is employed for all methods.

| METHOD | 1 TARGET | 2 TARGETS | 3 TARGETS |
|---|---|---|---|
| NEAR | **0.0382** | **0.0461** | **0.0860** |
| EMAC | 0.1454 | 0.1941 | 0.2503 |
| NEAR w/O R | 1.0399 | 1.0501 | 1.0308 |
| SIREN | 1.0077 | 1.0262 | 1.0543 |

**Angular Resolution.** The resolution probability (defined in Appendix C.1.2) of NEAR compared to baselines and the full *virtual* array benchmark is illustrated in Figure 2. Our method consistently achieves the highest resolution

probability among baselines and closely follows the full benchmark. While EMaC achieves comparable resolution probability for larger angle separations, its performance degrades significantly as the angle separation decreases. This is because convex relaxation techniques, such as the Nuclear Norm used in EMaC, impose separation conditions that inherently limit resolution (even in noise-free scenarios) (Dai & Milenkovic, 2009). In contrast, NEAR demonstrates robust resolution across different separations.

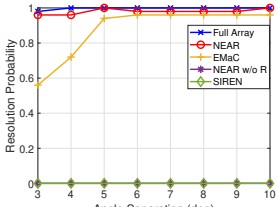
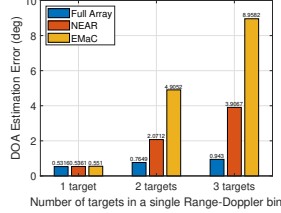

*Figure 2.* Angular resolution performance vs. angle separations.

*Figure 3.* DOA estimation accuracy vs. different number of targets.

**DOA Estimation.** The DOA estimation error of NEAR, compared to baselines and the full *virtual* array benchmark, is presented in Figure 3. Both NEAR and EMaC achieve similar performance to the benchmark when estimating the DOA of a single target. However, for multiple targets, NEAR significantly outperforms EMaC, demonstrating superior robustness in resolving closely spaced sources. Notably, EMaC's performance deteriorates as the number of targets increases, whereas NEAR maintains a lower estimation error, showing its capacity to generalize effectively to more complex scenarios.

**Ablation Study.** Tables 1 - 3 present a comprehensive ablation study assessing the impact of the physics-informed regularizer on NEAR. Without this regularizer, implicit neural representations (INRs) merely perform data fitting on the observed array responses but fail to capture the inherent low-rank structure in the Hankel matrix of the noiseless full *virtual* array response. This limitation severely affects the model's ability to generalize beyond observed data. The findings confirm that leveraging physics-informed constraints allows NEAR to achieve superior signal reconstruction and DOA estimation accuracy, particularly in challenging multi-target scenarios.

### 5.2. Real-world Experiments

We further conduct experiments using a commercial MIMO radar platform (IMAGEVK-74) as shown in Figure 5. IMAGEVK-74 employs 20 Tx antennas on a vertical line and 20 Rx antennas on a horizontal line, resulting in a *virtual* array of 20 × 20. IMAGEVK-74 transmits a Stepped-Frequency Continuous Wave (SFCW) waveform and the

bandwidth is set to be 67–69 GHz. The antenna spacing is roughly half of the wavelength. After collecting the 20 × 20 full array response matrix, we select a subset of data and treat it as a sparse set of measurements. Our procedure for active sensing using NEAR is depicted in Figure 4. Additional details on radar data processing (such as analog-to-digital conversion) across range and Doppler cells are included in the Appendix C.3.

**Angular resolution.** To measure the angular resolution, we put two corner reflectors at the boresight of the radar and gradually reduced the spacing between them. We employ the same signal processing pipeline (e.g., *beamforming*) and record the angular separation when the two targets merge in the radar angular spectrum. Table 4 shows the measured angular resolution with different setups. As the distance increases, the SNR reduces, and the reflected signal becomes weaker. NEAR achieves similar performance as the full array across all the range settings, *confirming its robustness at lower SNR conditions in real-world environments.*

*Table 4.* Smallest angular separation that can be resolved across various distance (SNR).

| METHOD | 2M | 3M | 4.5M |
|---|---|---|---|
| BENCHMARK | **5.7248°** | **6.6769°** | **6.9941°** |
| NEAR | **5.7248°** | **6.6769°** | **6.9941°** |
| EMAC | 8.5783° | 9.5273° | 10.1592° |
| NERF$^2$ | 8.5783° | 8.5783° | 8.8948° |

**Target localization.** We put several corner reflectors (1 – 4) in random positions in the field view of the radar and perform radar localization. The location of the reflectors spans 1 to 4 m in range, -45° to 45° in the azimuth angle, and -20° to 20° in the elevation angle. A total of 70 location samples are collected and their localization errors are calculated. Table 5 shows that NEAR outperforms the full array baseline, NeRF$^2$ and EMaC in terms of mean absolute error. NEAR exhibits a denoising effect that improves the localization accuracy compared with the full array baseline. This denoising was achieved by using only an upper bound (and not the exact value) on the number of targets to design the regularizer. The results further confirm *NEAR's capability to work in a complicated real-world environment with multiple reflectors*. See more experimental results in Appendix C.4.

**Computation Time.** Table 6 reports the averaged running times: NEAR (our approach) finishes in roughly 9 minutes, whereas EMaC and NeRF$^2$ require about 20 and 21 minutes, respectively. These results highlight the potential for real-time implementation of our approach with future advances in algorithms and computing hardware.

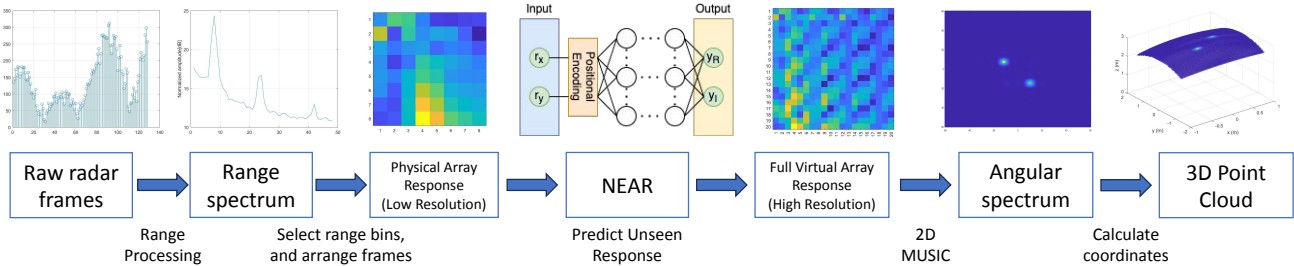

*Figure 4.* Radar active sensing workflow.

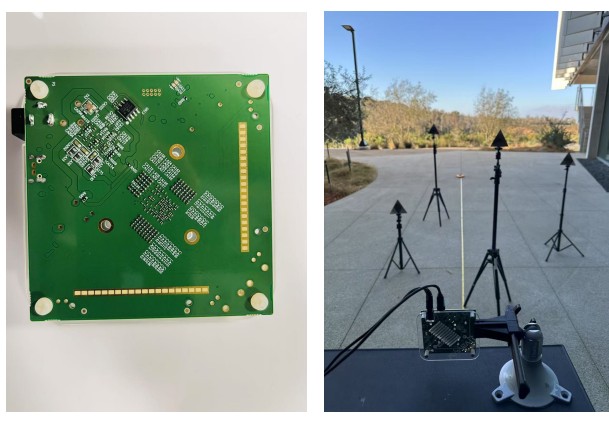

*Figure 5.* Left: 2D MIMO radar platform. Right: Real-world experimental setup.

*Table 5.* Localization accuracy for different number of targets ($\mathcal{K}$) in the environment.

| METHOD | $\mathcal{K} = 1$ | $\mathcal{K} = 2$ | $\mathcal{K} = 3$ | $\mathcal{K} = 4$ |
|---|---|---|---|---|
| BENCHMARK | 0.0827 | 0.0903 | 0.0965 | 0.0964 |
| NEAR | **0.0744** | **0.0770** | **0.0762** | **0.0718** |
| EMAC | 0.1062 | 0.1158 | 0.1170 | 0.1157 |
| NERF$^2$ | 0.4902 | 0.5096 | 0.4346 | 0.3898 |

## 6. Conclusions and Future Work

We proposed NEAR, the first framework that leverages Implicit Neural Representations to model and predict antenna array responses with sparse measurements without training data. By integrating harmonic signal structure and planar wave propagation models, NEAR effectively enables enhanced angular resolution and robustness in radar sensing applications. We believe NEAR represents the first step towards bridging the gap between deep learning-based neural fields and classical electromagnetic sensing and signal processing, unlocking new possibilities for super-resolution radar, wireless and autonomous sensing applications. Future work will focus on addressing the following challenges and improvements:

**Computational Efficiency.** Optimizing the framework for real-time inference on embedded radar hardware, reducing

*Table 6.* Computation time comparison.

| METHOD | NEAR | NERF$^2$ | EMAC |
|---|---|---|---|
| AVERAGED TIME COST (S) | **550.83** | 1278.31 | 1226.15 |

computational overhead while maintaining accuracy.

**Multi-Modal Sensor Fusion.** Integrating NEAR with Li-DAR, camera, and RF-based sensing to enhance robustness in complex environmental conditions.

## Acknowledgements

We thank the reviewers for their insightful comments. Additionally, we would like to thank Xingyu Chen for helpful discussions and instructions on implementation and experimentation. We also thank Parthasarathi Khirwadkar and Mohamed Hamdy for their valuable comments and feedback. This work is generously supported by the UC San Diego Center for Wireless Communications, by ONR N00014-19-1-2227, DOE DE-SC0022165, and by NSF under grants CNS-2128588, CNS-2312715, CNS-2403124, and NSF 2124929.

## Impact Statement

This paper presents work whose goal is to advance the field of Machine Learning. There are many potential societal consequences of our work, none which we feel must be specifically highlighted here.

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

# A. Proof of Theorem 4.1

## A.1. Notations and Definitions

Let $A, B \subseteq \mathbb{R}^n$ be two sets in $n$-dimensional Euclidean space. The Minkowski sum or difference of $A$ and $B$ is denoted by $A + B$, $A - B$ respectively, and defined as:

$$A + B = \{\boldsymbol{a} + \boldsymbol{b} \mid \boldsymbol{a} \in A, \ \boldsymbol{b} \in B\}, \quad A - B = \{\boldsymbol{a} - \boldsymbol{b} \mid \boldsymbol{a} \in A, \ \boldsymbol{b} \in B\}$$

Additionally, we define $\mathcal{D}(A, B)$ as the union of the Minkowski sum and difference of $A$ and $B$, given by:

$$\mathcal{D}(A, B) := (A + B) \cup (A - B).$$

We define $\mathcal{U}^{(q)}$ and $\mathcal{B}^{(q)}$ as follows:

$$\mathcal{U}^{(q)} = \left\{ \boldsymbol{s}^{(q)} = \left[ s_1^{(q)}, \cdots, s_T^{(q)} \right]^\top \ \Bigg| \ s_t^{(q)} \in \mathbb{Z}, \sum_{t=1}^T |s_t^{(q)}| \leq q \right\},$$

$$\mathcal{B}^{(q)} = \left\{ \boldsymbol{s}^{(q)} = \left[ s_1^{(q)}, \cdots, s_T^{(q)} \right]^\top \ \Bigg| \ s_t^{(q)} \in \mathbb{Z}, \sum_{t=1}^T |s_t^{(q)}| = q \right\}.$$

Recall that we are interested in analyzing the expressive power of INR architectures, which consist of a mapping function $\gamma : \mathbb{R}^D \to \mathbb{R}^T$ (positional encoding) followed by a multilayer perceptron (MLP). The MLP is parameterized by weights $\boldsymbol{W}^{(\ell)} \in \mathbb{R}^{F_\ell \times F_{\ell-1}}$, biases $\boldsymbol{b}^{(\ell)} \in \mathbb{R}^{F_\ell}$, and activation functions $\rho^{(\ell)} : \mathbb{R} \to \mathbb{R}$ applied elementwise at each layer $\ell = 1, \ldots, L-1$. Specifically, denoting the post-activation output of each layer as $\boldsymbol{z}^{(\ell)}$, most INR architectures compute:

$$\boldsymbol{z}^{(0)} = \gamma(\boldsymbol{r}),$$
$$\boldsymbol{z}^{(\ell)} = \rho^{(\ell)} \left( \boldsymbol{W}^{(\ell)} \boldsymbol{z}^{(\ell-1)} + \boldsymbol{b}^{(\ell)} \right), \quad \ell = 1, \ldots, L-1,$$
$$f_{\boldsymbol{\Theta}}(\boldsymbol{r}) = \boldsymbol{W}^{(L)} \boldsymbol{z}^{(L-1)} + \boldsymbol{b}^{(L)},$$

where $\boldsymbol{r} \in \mathbb{R}^D$ is the input coordinate. As it is plausible to normalize the inputs ($\boldsymbol{r}$) to their bounds, we assume that each variable's period is 1 (normalized to $[0, 1)$) or 2 (normalized to $[-1, 1)$).

## A.2. Lemma A.1 and Proof

**Lemma A.1.** *Let $\boldsymbol{\Omega} = [\boldsymbol{\omega}_1, \ldots, \boldsymbol{\omega}_T]^\top \in \mathbb{R}^{T \times D}$ be a frequency matrix, and let $\mathcal{S}_1, \mathcal{S}_2 \subseteq \mathbb{R}^T$ denote two sets of weights corresponding to these frequencies. Additionally, let $\{\phi_{\boldsymbol{s}_1} \in \mathbb{R} \mid \boldsymbol{s}_1 \in \mathcal{S}_1\}$ and $\{\phi_{\boldsymbol{s}_2} \in \mathbb{R} \mid \boldsymbol{s}_2 \in \mathcal{S}_2\}$ represent two collections of scalar phases, and $\{\beta_{\boldsymbol{s}_1} \in \mathbb{R} \mid \boldsymbol{s}_1 \in \mathcal{S}_1\}$ and $\{\beta_{\boldsymbol{s}_2} \in \mathbb{R} \mid \boldsymbol{s}_2 \in \mathcal{S}_2\}$ two corresponding sets of scalar coefficients. For any $\boldsymbol{r} \in \mathbb{R}^D$, the following holds:*

$$\left( \sum_{\boldsymbol{s}_1 \in \mathcal{S}_1} \beta_{\boldsymbol{s}_1} \cos \left( \langle \boldsymbol{\Omega}^\top \boldsymbol{s}_1, \boldsymbol{r} \rangle + \phi_{\boldsymbol{s}_1} \right) \right) \left( \sum_{\boldsymbol{s}_2 \in \mathcal{S}_2} \beta_{\boldsymbol{s}_2} \cos \left( \langle \boldsymbol{\Omega}^\top \boldsymbol{s}_2, \boldsymbol{r} \rangle + \phi_{\boldsymbol{s}_2} \right) \right) = \sum_{\boldsymbol{s}' \in \mathcal{D}(\mathcal{S}_1, \mathcal{S}_2)} \tilde{\beta}_{\boldsymbol{s}'} \cos \left( \langle \boldsymbol{\Omega}^\top \boldsymbol{s}', \boldsymbol{r} \rangle + \tilde{\phi}_{\boldsymbol{s}'} \right),$$

$$\tag{12}$$

*where*

$$\mathcal{D}(\mathcal{S}_1, \mathcal{S}_2) = (\mathcal{S}_1 + \mathcal{S}_2) \cup (\mathcal{S}_1 - \mathcal{S}_2), \tag{13}$$

*and $\{\tilde{\phi}_{\boldsymbol{s}'} \in \mathbb{R} \mid \boldsymbol{s}' \in \mathcal{D}(\mathcal{S}_1, \mathcal{S}_2)\}$ and $\{\tilde{\beta}_{\boldsymbol{s}'} \in \mathbb{R} \mid \boldsymbol{s}' \in \mathcal{D}(\mathcal{S}_1, \mathcal{S}_2)\}$ denote the resulting scalar phases and coefficients, respectively.*

*Proof.*

$$\left( \sum_{\boldsymbol{s}_1 \in \mathcal{S}_1} \beta_{\boldsymbol{s}_1} \cos \left( \langle \boldsymbol{\Omega}^\top \boldsymbol{s}_1, \boldsymbol{r} \rangle + \phi_{\boldsymbol{s}_1} \right) \right) \left( \sum_{\boldsymbol{s}_2 \in \mathcal{S}_2} \beta_{\boldsymbol{s}_2} \cos \left( \langle \boldsymbol{\Omega}^\top \boldsymbol{s}_2, \boldsymbol{r} \rangle + \phi_{\boldsymbol{s}_2} \right) \right)$$

$$= \sum_{\boldsymbol{s}_1 \in \mathcal{S}_1} \sum_{\boldsymbol{s}_2 \in \mathcal{S}_2} \beta_{\boldsymbol{s}_1} \beta_{\boldsymbol{s}_2} \cos \left( \langle \boldsymbol{\Omega}^\top \boldsymbol{s}_1, \boldsymbol{r} \rangle + \phi_{\boldsymbol{s}_1} \right) \cos \left( \langle \boldsymbol{\Omega}^\top \boldsymbol{s}_2, \boldsymbol{r} \rangle + \phi_{\boldsymbol{s}_2} \right)$$

$$= \sum_{\boldsymbol{s}_1 \in \mathcal{S}_1} \sum_{\boldsymbol{s}_2 \in \mathcal{S}_2} \beta_{\boldsymbol{s}_1} \beta_{\boldsymbol{s}_2} \frac{1}{2} \left( \cos \left( \langle \boldsymbol{\Omega}^\top (\boldsymbol{s}_1 + \boldsymbol{s}_2), \boldsymbol{r} \rangle + \phi_{\boldsymbol{s}_1} + \phi_{\boldsymbol{s}_2} \right) + \cos \left( \langle \boldsymbol{\Omega}^\top (\boldsymbol{s}_1 - \boldsymbol{s}_2), \boldsymbol{r} \rangle + \phi_{\boldsymbol{s}_1} - \phi_{\boldsymbol{s}_2} \right) \right)$$

$$= \sum_{\boldsymbol{s}' \in \mathcal{D}(\mathcal{S}_1, \mathcal{S}_2)} \tilde{\beta}_{\boldsymbol{s}'} \cos \left( \langle \boldsymbol{\Omega}^\top \boldsymbol{s}', \boldsymbol{r} \rangle + \tilde{\phi}_{\boldsymbol{s}'} \right).$$

The last equality combines terms with the same frequency, where $\tilde{\beta}_{\boldsymbol{s}'}$ and $\tilde{\phi}_{\boldsymbol{s}'}$ represent the resultant magnitude and phase, respectively, obtained through phasor addition after grouping. This technique will be used repeatedly in the following subsections to simplify analogous expressions. □

### A.3. Lemma A.2 and Proof

**Lemma A.2.** *Let $\boldsymbol{\Omega} = [\boldsymbol{\omega}_1, \ldots, \boldsymbol{\omega}_T]^\top \in \mathbb{R}^{T \times D}$ be a frequency matrix, and let $E_T = \{ \boldsymbol{e}_t \in \mathbb{R}^T \mid t \in \mathbb{Z}, 1 \le t \le T \}$ denote the set of canonical basis vectors of $\mathbb{R}^T$. Define $\mathcal{S}^{(1)}$ as the augmented set of $E_T$, given by $\mathcal{S}^{(1)} = \{ \boldsymbol{e}_t, -\boldsymbol{e}_t \mid \boldsymbol{e}_t \in E_T \}$. Additionally, let $\{ \phi_{\boldsymbol{s}^{(1)}} \in \mathbb{R} \mid \boldsymbol{s}^{(1)} \in \mathcal{S}^{(1)} \}$ be a collection of scalar phases, and $\{ \beta_{\boldsymbol{s}^{(1)}} \in \mathbb{R} \mid \boldsymbol{s}^{(1)} \in \mathcal{S}^{(1)} \}$ the corresponding set of scalar coefficients. For any $\boldsymbol{r} \in \mathbb{R}^D$ and $q \in \mathbb{N}$, the following equality holds:*

$$\left( \sum_{\boldsymbol{s}^{(1)} \in \mathcal{S}^{(1)}} \beta_{\boldsymbol{s}^{(1)}} \cos \left( \langle \boldsymbol{\Omega}^\top \boldsymbol{s}^{(1)}, \boldsymbol{r} \rangle + \phi_{\boldsymbol{s}^{(1)}} \right) \right)^q = \sum_{\boldsymbol{s}^{(q)} \in \mathcal{S}^{(q)}} \tilde{\beta}_{\boldsymbol{s}^{(q)}} \cos \left( \langle \boldsymbol{\Omega}^\top \boldsymbol{s}^{(q)}, \boldsymbol{r} \rangle + \tilde{\phi}_{\boldsymbol{s}^{(q)}} \right), \tag{14}$$

*where*

$$\mathcal{S}^{(q)} := \mathcal{D}(\mathcal{S}^{(q-1)}, \mathcal{S}^{(1)}), \quad \mathcal{B}^{(q)} \subseteq \mathcal{S}^{(q)} \subseteq \mathcal{U}^{(q)}, \tag{15}$$

*for some $\{ \tilde{\phi}_{\boldsymbol{s}^{(q)}} \in \mathbb{R} \mid \boldsymbol{s}^{(q)} \in \mathcal{S}^{(q)} \}$ and $\{ \tilde{\beta}_{\boldsymbol{s}^{(q)}} \in \mathbb{R} \mid \boldsymbol{s}^{(q)} \in \mathcal{S}^{(q)} \}$.*

*Proof.* We will use induction to prove our statement. The statement trivially holds for $q = 1$ since $\mathcal{B}^{(1)} = \mathcal{S}^{(1)} \subseteq \mathcal{U}^{(1)}$. Assume it also holds for $q > 1$, we first show that $\mathcal{S}^{(q+1)} \subseteq \mathcal{U}^{(q+1)}$. Using the induction hypothesis and Lemma A.1, we have:

$$\left( \sum_{\boldsymbol{s}^{(1)} \in \mathcal{S}^{(1)}} \beta_{\boldsymbol{s}^{(1)}} \cos \left( \langle \boldsymbol{\Omega}^\top \boldsymbol{s}^{(1)}, \boldsymbol{r} \rangle + \phi_{\boldsymbol{s}^{(1)}} \right) \right)^{q+1}$$

$$= \left( \sum_{\boldsymbol{s}^{(1)} \in \mathcal{S}^{(1)}} \beta_{\boldsymbol{s}^{(1)}} \cos \left( \langle \boldsymbol{\Omega}^\top \boldsymbol{s}^{(1)}, \boldsymbol{r} \rangle + \phi_{\boldsymbol{s}^{(1)}} \right) \right)^q \left( \sum_{\boldsymbol{s}^{(1)} \in \mathcal{S}^{(1)}} \beta_{\boldsymbol{s}^{(1)}} \cos \left( \langle \boldsymbol{\Omega}^\top \boldsymbol{s}^{(1)}, \boldsymbol{r} \rangle + \phi_{\boldsymbol{s}^{(1)}} \right) \right)$$

$$= \left( \sum_{\boldsymbol{s}^{(q)} \in \mathcal{S}^{(q)}} \tilde{\beta}_{\boldsymbol{s}^{(q)}} \cos \left( \langle \boldsymbol{\Omega}^\top \boldsymbol{s}^{(q)}, \boldsymbol{r} \rangle + \tilde{\phi}_{\boldsymbol{s}^{(q)}} \right) \right) \left( \sum_{\boldsymbol{s}^{(1)} \in \mathcal{S}^{(1)}} \beta_{\boldsymbol{s}^{(1)}} \cos \left( \langle \boldsymbol{\Omega}^\top \boldsymbol{s}^{(1)}, \boldsymbol{r} \rangle + \phi_{\boldsymbol{s}^{(1)}} \right) \right)$$

$$= \sum_{\boldsymbol{s}^{(q+1)} \in \mathcal{D}(\mathcal{S}^{(q)}, \mathcal{S}^{(1)})} \beta'_{\boldsymbol{s}^{(q+1)}} \cos \left( \langle \boldsymbol{\Omega}^\top \boldsymbol{s}^{(q+1)}, \boldsymbol{r} \rangle + \phi'_{\boldsymbol{s}^{(q+1)}} \right)$$

$$= \sum_{\boldsymbol{s}^{(q+1)} \in \mathcal{S}^{(q+1)}} \beta'_{\boldsymbol{s}^{(q+1)}} \cos \left( \langle \boldsymbol{\Omega}^\top \boldsymbol{s}^{(q+1)}, \boldsymbol{r} \rangle + \phi'_{\boldsymbol{s}^{(q+1)}} \right)$$

Moreover,

$$
\begin{aligned}
\mathcal{S}^{(q+1)} = \mathcal{D}\{\mathcal{S}^{(q)}, \mathcal{S}^{(1)}\} &= \left\{ \boldsymbol{s}^{(q+1)} = \boldsymbol{s}^{(q)} \pm \boldsymbol{s}^{(1)} \;\middle|\; \boldsymbol{s}^{(q)} \in \mathcal{S}^{(q)}, \boldsymbol{s}^{(1)} \in \mathcal{S}^{(1)} \right\} \\
&\subseteq \left\{ \boldsymbol{s}^{(q+1)} = \boldsymbol{s}^{(q)} \pm \boldsymbol{s}^{(1)} \;\middle|\; \boldsymbol{s}^{(q)} \in \mathcal{U}^{(q)}, \boldsymbol{s}^{(1)} \in \mathcal{S}^{(1)} \right\} \\
&= \left\{ \boldsymbol{s}^{(q+1)} = \left[ s_1^{(q)}, \cdots, s_T^{(q)} \right]^\top \pm \boldsymbol{e}_t \;\middle|\; s_t^{(q)} \in \mathbb{Z}, \sum_{t=1}^{T} |s_t^{(q)}| \leq q, \boldsymbol{e}_t \in \mathcal{S}^{(1)} \right\} \\
&\subseteq \left\{ \boldsymbol{s}^{(q+1)} = \left[ s_1^{(q+1)}, \cdots, s_T^{(q+1)} \right]^\top \;\middle|\; s_t^{(q+1)} \in \mathbb{Z}, \sum_{t=1}^{T} |s_t^{(q+1)}| \leq q+1 \right\} = \mathcal{U}^{(q+1)},
\end{aligned}
$$

where the last line follows from triangle inequality. Next, we show that given the assumption, we have $\mathcal{B}^{(q+1)} \subseteq \mathcal{S}^{(q+1)}$:

$$
\begin{aligned}
\mathcal{S}^{(q+1)} = \mathcal{D}\{\mathcal{S}^{(q)}, \mathcal{S}^{(1)}\} &= \left\{ \boldsymbol{s}^{(q+1)} = \boldsymbol{s}^{(q)} \pm \boldsymbol{s}^{(1)} \;\middle|\; \boldsymbol{s}^{(q)} \in \mathcal{S}^{(q)}, \boldsymbol{s}^{(1)} \in \mathcal{S}^{(1)} \right\} \\
&\supseteq \left\{ \boldsymbol{s}^{(q+1)} = \boldsymbol{s}^{(q)} \pm \boldsymbol{s}^{(1)} \;\middle|\; \boldsymbol{s}^{(q)} \in \mathcal{B}^{(q)}, \boldsymbol{s}^{(1)} \in \mathcal{S}^{(1)} \right\} \\
&= \left\{ \boldsymbol{s}^{(q+1)} = \left[ s_1^{(q)}, \cdots, s_T^{(q)} \right]^\top \pm \boldsymbol{e}_t \;\middle|\; s_t^{(q)} \in \mathbb{Z}, \sum_{t=1}^{T} |s_t^{(q)}| = q, \boldsymbol{e}_t \in \mathcal{S}^{(1)} \right\} \\
&\supseteq \left\{ \boldsymbol{s}^{(q+1)} = \left[ s_1^{(q+1)}, \cdots, s_T^{(q+1)} \right]^\top \;\middle|\; s_t^{(q+1)} \in \mathbb{Z}, \sum_{t=1}^{T} |s_t^{(q+1)}| = q+1 \right\} = \mathcal{B}^{(q+1)}.
\end{aligned}
$$

Therefore we have $\mathcal{B}^{(q+1)} \subseteq \mathcal{S}^{(q+1)} \subseteq \mathcal{U}^{(q+1)}$. Thus by induction (14) holds $\forall\, q \in \mathbb{N}$. $\qquad\square$

## A.4. Lemma A.3 and Proof

**Lemma A.3.** *Let* $\boldsymbol{\Omega} = [\boldsymbol{\omega}_1, \ldots, \boldsymbol{\omega}_T]^\top \in \mathbb{R}^{T \times D}$ *be a frequency matrix, and let* $E_T = \{\boldsymbol{e}_t \in \mathbb{R}^T \mid t \in \mathbb{Z}, 1 \leq t \leq T\}$ *denote the set of canonical basis vectors of* $\mathbb{R}^T$. *Define* $\mathcal{S}^{(1)}$ *as the augmented set of* $E_T$, *given by* $\mathcal{S}^{(1)} = \{\boldsymbol{e}_t, -\boldsymbol{e}_t \mid \boldsymbol{e}_t \in E_T\}$. *Additionally, let* $\{\phi_{\boldsymbol{s}^{(1)}} \in \mathbb{R} \mid \boldsymbol{s}^{(1)} \in \mathcal{S}^{(1)}\}$ *be a collection of scalar phases, and* $\{\beta_{\boldsymbol{s}^{(1)}} \in \mathbb{R} \mid \boldsymbol{s}^{(1)} \in \mathcal{S}^{(1)}\}$ *the corresponding set of scalar coefficients. For any* $\boldsymbol{r} \in \mathbb{R}^D$, $Q \in \mathbb{N}$ *and* $\alpha_q \in \mathbb{R}$ $(q = 1, \cdots, Q)$, *the following equality holds:*

$$
\sum_{q=0}^{Q} \alpha_q \left( \sum_{\boldsymbol{s}^{(1)} \in \mathcal{S}^{(1)}} \beta_{\boldsymbol{s}^{(1)}} \cos\left( \langle \boldsymbol{\Omega}^\top \boldsymbol{s}^{(1)}, \boldsymbol{r} \rangle + \phi_{\boldsymbol{s}^{(1)}} \right) \right)^q = \sum_{\bar{\boldsymbol{s}}^{(Q)} \in \bar{\mathcal{S}}^{(Q)}} \tilde{\tilde{\beta}}_{\bar{\boldsymbol{s}}^{(Q)}} \cos\left( \langle \boldsymbol{\Omega}^\top \bar{\boldsymbol{s}}^{(Q)}, \boldsymbol{r} \rangle + \tilde{\tilde{\phi}}_{\bar{\boldsymbol{s}}^{(Q)}} \right), \tag{16}
$$

*where*

$$
\bar{\mathcal{S}}^{(Q)} = \bigcup_{q=1}^{Q} \mathcal{S}^{(q)} = \left\{ \bar{\boldsymbol{s}}^{(Q)} = \left[ \bar{s}_1^{(Q)}, \cdots, \bar{s}_T^{(Q)} \right]^\top \;\middle|\; \bar{s}_t^{(Q)} \in \mathbb{Z} \wedge \sum_{t=1}^{T} |\bar{s}_t^{(Q)}| \leq Q \right\} = \mathcal{U}^{(Q)} \tag{17}
$$

*for some* $\{\tilde{\tilde{\phi}}_{\bar{\boldsymbol{s}}^{(Q)}} \in \mathbb{R} \mid \bar{\boldsymbol{s}}^{(Q)} \in \bar{\mathcal{S}}^{(Q)}\}$ *and* $\{\tilde{\tilde{\beta}}_{\bar{\boldsymbol{s}}^{(Q)}} \in \mathbb{R} \mid \bar{\boldsymbol{s}}^{(Q)} \in \bar{\mathcal{S}}^{(Q)}\}$.

*Proof.* According to Lemma A.2, we have:

$$
\begin{aligned}
\sum_{q=0}^{Q} \alpha_q \left( \sum_{\boldsymbol{s}^{(1)} \in \mathcal{S}^{(1)}} \beta_{\boldsymbol{s}^{(1)}} \cos\left( \langle \boldsymbol{\Omega}^\top \boldsymbol{s}^{(1)}, \boldsymbol{r} \rangle + \phi_{\boldsymbol{s}^{(1)}} \right) \right)^q &= \sum_{q=0}^{Q} \alpha_q \sum_{\boldsymbol{s}^{(q)} \in \mathcal{S}^{(q)}} \tilde{\beta}_{\boldsymbol{s}^{(q)}} \cos\left( \langle \boldsymbol{\Omega}^\top \boldsymbol{s}^{(q)}, \boldsymbol{r} \rangle + \tilde{\phi}_{\boldsymbol{s}^{(q)}} \right) \\
&= \sum_{\bar{\boldsymbol{s}}^{(Q)} \in \bar{\mathcal{S}}^{(Q)}} \tilde{\tilde{\beta}}_{\bar{\boldsymbol{s}}^{(Q)}} \cos\left( \langle \boldsymbol{\Omega}^\top \bar{\boldsymbol{s}}^{(Q)}, \boldsymbol{r} \rangle + \tilde{\tilde{\phi}}_{\bar{\boldsymbol{s}}^{(Q)}} \right),
\end{aligned}
$$

where $\bar{\mathcal{S}}^{(Q)} = \bigcup_{q=1}^{Q} \mathcal{S}^{(q)}$. Since $\mathcal{B}^{(q)} \subseteq \mathcal{S}^{(q)} \subseteq \mathcal{U}^{(q)}$ for $q \in \mathbb{N}$, we then have:

$$
\bigcup_{q=1}^{Q} \mathcal{B}^{(q)} \subseteq \bar{\mathcal{S}}^{(Q)} \subseteq \bigcup_{q=1}^{Q} \mathcal{U}^{(q)}.
$$

According to the definition of $\mathcal{B}^{(q)}$ and $\mathcal{U}^{(q)}$, we have:

$$
\begin{aligned}
\bigcup_{q=1}^{Q} \mathcal{B}^{(q)} &= \bigcup_{q=1}^{Q} \left\{ \boldsymbol{s}^{(q)} = \left[s_1^{(q)}, \cdots, s_T^{(q)}\right]^{\top} \;\middle|\; s_t^{(q)} \in \mathbb{Z}, \sum_{t=1}^{T} |s_t^{(q)}| = q \right\} \\
&= \left\{ \boldsymbol{s}^{(Q)} = \left[s_1^{(Q)}, \cdots, s_T^{(Q)}\right]^{\top} \;\middle|\; s_t^{(Q)} \in \mathbb{Z}, \sum_{t=1}^{T} |s_t^{(Q)}| \in \{1, 2, \cdots, Q\} \right\}; \\
\bigcup_{q=1}^{Q} \mathcal{U}^{(q)} &= \bigcup_{q=1}^{Q} \left\{ \boldsymbol{s}^{(q)} = \left[s_1^{(q)}, \cdots, s_T^{(q)}\right]^{\top} \;\middle|\; s_t^{(q)} \in \mathbb{Z}, \sum_{t=1}^{T} |s_t^{(q)}| \leq q \right\} \\
&= \left\{ \boldsymbol{s}^{(q)} = \left[s_1^{(q)}, \cdots, s_T^{(q)}\right]^{\top} \;\middle|\; s_t^{(q)} \in \mathbb{Z}, \sum_{t=1}^{T} |s_t^{(q)}| \leq Q \right\} = \mathcal{U}^{(Q)}.
\end{aligned}
$$

Moreover, $\bar{\mathcal{S}}^{(Q)} \supset \mathcal{B}^{(0)} = \left\{ \boldsymbol{s}^{(0)} = \left[s_1^{(0)}, \cdots, s_T^{(0)}\right]^{\top} \;\middle|\; s_t^{(0)} \in \mathbb{Z}, \sum_{t=1}^{T} |s_t^{(0)}| = 0 \right\}$ since it is easy to verify that $\mathcal{S}^{(2)} \supset \mathcal{B}^{(0)}$. Therefore, we have

$$
\mathcal{U}^{(Q)} = \bigcup_{q=1}^{Q} \mathcal{B}^{(q)} \cup \mathcal{B}^{(0)} \subseteq \bar{\mathcal{S}}^{(Q)} \subseteq \mathcal{U}^{(Q)} \implies \bar{\mathcal{S}}^{(Q)} = \mathcal{U}^{(Q)}.
$$

$\square$

### A.5. Lemma A.4 and Proof

**Lemma A.4.** *Let $\boldsymbol{\Omega} = [\boldsymbol{\omega}_1, \ldots, \boldsymbol{\omega}_T]^{\top} \in \mathbb{R}^{T \times D}$ be a frequency matrix, and let $E_T = \{\boldsymbol{e}_t \in \mathbb{R}^T \mid t \in \mathbb{Z}, 1 \leq t \leq T\}$ denote the set of canonical basis vectors of $\mathbb{R}^T$. Define $\mathcal{S}^{(1)}$ as the augmented set of $E_T$, given by $\mathcal{S}^{(1)} = \{\boldsymbol{e}_t, -\boldsymbol{e}_t \mid \boldsymbol{e}_t \in E_T\}$. Additionally, let $\{\phi_{\boldsymbol{s}^{(1)}} \in \mathbb{R} \mid \boldsymbol{s}^{(1)} \in \mathcal{S}^{(1)}\}$ be a collection of scalar phases, and $\{\beta_{\boldsymbol{s}^{(1)}} \in \mathbb{R} \mid \boldsymbol{s}^{(1)} \in \mathcal{S}^{(1)}\}$ the corresponding set of scalar coefficients. For any $\boldsymbol{r} \in \mathbb{R}^D$ and $q, p \in \mathbb{N}$, the following equality holds:*

$$
\left( \sum_{\boldsymbol{s}^{(p)} \in \mathcal{U}^{(p)}} \beta_{\boldsymbol{s}^{(p)}} \cos\left(\langle \boldsymbol{\Omega}^{\top} \boldsymbol{s}^{(p)}, \boldsymbol{r} \rangle + \phi_{\boldsymbol{s}^{(p)}}\right) \right)^{q} = \left( \sum_{\boldsymbol{s}^{(qp)} \in \mathcal{U}^{(qp)}} \tilde{\beta}_{\boldsymbol{s}^{(qp)}} \cos\left(\langle \boldsymbol{\Omega}^{\top} \boldsymbol{s}^{(qp)}, \boldsymbol{r} \rangle + \tilde{\phi}_{\boldsymbol{s}^{(qp)}}\right) \right), \tag{18}
$$

*for some $\{\tilde{\beta}_{\boldsymbol{s}^{(qp)}} \in \mathbb{R} \mid \boldsymbol{s}^{(qp)} \in \mathcal{U}^{(qp)}\}$ and $\{\tilde{\phi}_{\boldsymbol{s}^{(qp)}} \in \mathbb{R} \mid \boldsymbol{s}^{(qp)} \in \mathcal{U}^{(qp)}\}$.*

*Proof.* Again we will use induction to prove the statement. The statement trivially holds for $q = 1$. Assume it also holds for $q > 1$, then

$$
\begin{aligned}
&\left( \sum_{\boldsymbol{s}^{(p)} \in \mathcal{U}^{(p)}} \beta_{\boldsymbol{s}^{(p)}} \cos\left(\langle \boldsymbol{\Omega}^{\top} \boldsymbol{s}^{(p)}, \boldsymbol{r} \rangle + \phi_{\boldsymbol{s}^{(p)}}\right) \right)^{q+1} \\
&= \left( \sum_{\boldsymbol{s}^{(p)} \in \mathcal{U}^{(p)}} \beta_{\boldsymbol{s}^{(p)}} \cos\left(\langle \boldsymbol{\Omega}^{\top} \boldsymbol{s}^{(p)}, \boldsymbol{r} \rangle + \phi_{\boldsymbol{s}^{(p)}}\right) \right)^{q} \left( \sum_{\boldsymbol{s}^{(p)} \in \mathcal{U}^{(p)}} \beta_{\boldsymbol{s}^{(p)}} \cos\left(\langle \boldsymbol{\Omega}^{\top} \boldsymbol{s}^{(p)}, \boldsymbol{r} \rangle + \phi_{\boldsymbol{s}^{(p)}}\right) \right) \\
&= \left( \sum_{\boldsymbol{s}^{(qp)} \in \mathcal{U}^{(qp)}} \beta_{\boldsymbol{s}^{(qp)}} \cos\left(\langle \boldsymbol{\Omega}^{\top} \boldsymbol{s}^{(qp)}, \boldsymbol{r} \rangle + \phi_{\boldsymbol{s}^{(qp)}}\right) \right) \left( \sum_{\boldsymbol{s}^{(p)} \in \mathcal{U}^{(p)}} \beta_{\boldsymbol{s}^{(p)}} \cos\left(\langle \boldsymbol{\Omega}^{\top} \boldsymbol{s}^{(p)}, \boldsymbol{r} \rangle + \phi_{\boldsymbol{s}^{(p)}}\right) \right) \\
&= \left( \sum_{\boldsymbol{s}' \in \mathcal{D}\left(\mathcal{U}^{(qp)}, \mathcal{U}^{(p)}\right)} \tilde{\beta}_{\boldsymbol{s}'} \cos\left(\langle \boldsymbol{\Omega}^{\top} \boldsymbol{s}', \boldsymbol{r} \rangle + \tilde{\phi}_{\boldsymbol{s}'}\right) \right),
\end{aligned}
$$

where the second equality holds by assumption, and the last equality holds by Lemma A.1. Moreover, we have

$$
\begin{aligned}
\mathcal{D}\{\mathcal{U}^{(qp)}, \mathcal{U}^{(p)}\} &= \left\{ \boldsymbol{s}' = \boldsymbol{s}^{(qp)} + \boldsymbol{s}^{(p)} \;\middle|\; \boldsymbol{s}^{(qp)} \in \mathcal{U}^{(qp)}, \boldsymbol{s}^{(p)} \in \mathcal{U}^{(p)} \right\} \\
&= \left\{ \boldsymbol{s}' = \left[s_t^{(qp)}\right]_{1 \leq t \leq T}^{\top} + \left[s_t^{(p)}\right]_{1 \leq t \leq T}^{\top} \;\middle|\; s_t^{(qp)}, s_t^{(p)} \in \mathbb{Z} \wedge \sum_{t=1}^{T} |s_t^{(qp)}| \leq qp \wedge \sum_{t=1}^{T} |s_t^{(p)}| \leq p \right\} \\
&\subseteq \left\{ \boldsymbol{s}' = [s_t']_{1 \leq t \leq T}^{\top} \;\middle|\; s_t' \in \mathbb{Z} \wedge \sum_{t=1}^{T} |s_t'| \leq (q+1)p \right\} \\
&= \mathcal{U}^{((q+1)p)},
\end{aligned}
$$

where the third line follows from triangle inequality. Next, we will show that $\mathcal{U}^{((q+1)p)} \subseteq \mathcal{D}\{\mathcal{U}^{(qp)}, \mathcal{U}^{(p)}\}$. For $\boldsymbol{u} = [u_1, \cdots, u_T]^{\top} \in \mathcal{U}^{((q+1)p)}$, we would like to construct two vectors such that $\boldsymbol{v} = [v_1, \cdots, v_T]^{\top} \in \mathcal{U}^{(qp)}$, $\boldsymbol{w} = [w_1, \cdots, w_T]^{\top} \in \mathcal{U}^{(p)}$, and $\boldsymbol{u} = \boldsymbol{v} + \boldsymbol{w}$.

Let $\tilde{u} = \sum_{t=1}^{T} |u_t|$ and $\tilde{v} = \min(\tilde{u}, qp)$. It is easy to see $\tilde{u} \leq (q+1)p$ by definition. Suppose $t' \leq T$ is the largest integer that satisfies $\sum_{t=1}^{t'} |u_t| \leq \tilde{v}$. If $t' = T$, it follows that $\boldsymbol{v} = \boldsymbol{u}, \boldsymbol{w} = \boldsymbol{0}$, and hence the statement holds. If $t' < T$, we can argue that $|u_{t'+1}| > \tilde{v} - \sum_{t=1}^{t'} |u_t| \triangleq u'$, otherwise $\sum_{t=1}^{t'+1} |u_t| \leq \tilde{v}$ contradicts the assumption that $t' \leq T$ is the largest integer that satisfies $\sum_{t=1}^{t'} |u_t| \leq \tilde{v}$. Thus, we can construct such $\boldsymbol{v}$ and $\boldsymbol{w}$ as:

$$
\begin{aligned}
v_t &= \mathbf{1}_{\{t \leq t'\}} u_t + \mathbf{1}_{\{t = t'+1\}} \operatorname{sgn}(u_{t'+1}) u' \quad \forall t \in \mathbb{N}_+, t \leq T \\
w_t &= \mathbf{1}_{\{t = t'+1\}} \operatorname{sgn}(u_{t'+1})(|u_{t'+1}| - u') + \mathbf{1}_{\{t > t'+1\}} u_t \quad \forall t \in \mathbb{N}_+, t \leq T,
\end{aligned}
$$

where $\mathbf{1}_{\{\cdot\}}$ denotes the indicator function and $\operatorname{sgn}(\cdot)$ represents the sign function. It is easy to follow that $\tilde{v} = \sum_{t=1}^{T} |v_t|$ and $\tilde{u} = \tilde{v} + \sum_{t=1}^{T} |w_t|$. Then we will verify that $\boldsymbol{v} \in \mathcal{U}^{(qp)}$ and $\boldsymbol{w} \in \mathcal{U}^{(p)}$. By construction, $\sum_{t=1}^{T} |v_t| = \sum_{t=1}^{t'} |u_t| + u' = \tilde{v} \leq \min(\tilde{u}, qp) \leq qp$, and $\sum_{t=1}^{T} |w_t| = \tilde{u} - \tilde{v} = \tilde{u} - \min(\tilde{u}, qp) = \max(0, \tilde{u} - qp) \leq p$ since $\tilde{u} \leq (q+1)p$, which completes the construction rule. Therefore, we have $\mathcal{U}^{((q+1)p)} \subseteq \mathcal{D}\{\mathcal{U}^{(qp)}, \mathcal{U}^{(p)}\}$, and hence $\mathcal{U}^{((q+1)p)} = \mathcal{D}\{\mathcal{U}^{(qp)}, \mathcal{U}^{(p)}\}$. $\qquad\square$

## A.6. Main Proof of Theorem 4.1

*Proof.* To begin with the proof of Theorem 4.1, we will prove the following two statements first.

Define $\tilde{\boldsymbol{z}}^{(1)} = \boldsymbol{W}^{(1)} \boldsymbol{z}^{(0)} + \boldsymbol{b}^{(1)}$ as the pre-activation output of the first layer, where $\boldsymbol{z}^{(0)} = \gamma(\boldsymbol{r}) = \sin(\boldsymbol{\Omega}_{\mathcal{T}} \boldsymbol{r} + \boldsymbol{\phi}_{\mathcal{T}})$. Let $\{\phi_{\boldsymbol{s}^{(1)}} \in \mathbb{R} \mid \boldsymbol{s}^{(1)} \in \mathcal{S}^{(1)}\}$ be a collection of scalar phases, and $\{\beta_{\boldsymbol{s}^{(1)}} \in \mathbb{R} \mid \boldsymbol{s}^{(1)} \in \mathcal{S}^{(1)}\}$ the corresponding set of scalar coefficients. Let $\tilde{z}_i^{(1)}$ and $b_i^{(1)}$ denote the $i^{\text{th}}$ entries of $\tilde{\boldsymbol{z}}^{(1)}$ and $\boldsymbol{b}^{(1)}$, respectively, and let $\boldsymbol{W}_i^{(1)}$ represent the $i^{\text{th}}$ row of $\boldsymbol{W}^{(1)}$. We first would like to show that given $\{\phi_{\boldsymbol{s}^{(1)}} \in \mathbb{R} \mid \boldsymbol{s}^{(1)} \in \mathcal{S}^{(1)}\}$ and $\{\beta_{\boldsymbol{s}^{(1)}} \in \mathbb{R} \mid \boldsymbol{s}^{(1)} \in \mathcal{S}^{(1)}\}$ ($\mathcal{S}^{(1)}$ is defined in Lemma A.2):

$$
\sum_{\boldsymbol{s}^{(1)} \in \mathcal{S}^{(1)}} \beta_{\boldsymbol{s}^{(1)}} \cos\left( \langle \boldsymbol{\Omega}_{\mathcal{T}}^{\top} \boldsymbol{s}^{(1)}, \boldsymbol{r} \rangle + \phi_{\boldsymbol{s}^{(1)}} \right) + \zeta = \tilde{z}_i^{(1)} = \boldsymbol{W}_i^{(1)} \sin(\boldsymbol{\Omega}_{\mathcal{T}} \boldsymbol{r} + \boldsymbol{\phi}_{\mathcal{T}}) + b_i^{(1)}
$$

for some $\boldsymbol{W}_i^{(1)} \in \mathbb{R}^{1 \times T}$ and $b_i^{(1)} \in \mathbb{R}$. Note that adding constant does not affect frequency and interchanging sines with cosines only affects the phase terms. We can express the summation as follows:

$$
\begin{aligned}
\sum_{\boldsymbol{s}^{(1)} \in \mathcal{S}^{(1)}} \beta_{\boldsymbol{s}^{(1)}} \cos\left( \langle \boldsymbol{\Omega}_{\mathcal{T}}^{\top} \boldsymbol{s}^{(1)}, \boldsymbol{r} \rangle + \phi_{\boldsymbol{s}^{(1)}} \right) &= \sum_{t=1}^{T} \beta_t \cos(\langle \boldsymbol{\omega}_t, \boldsymbol{r} \rangle + \phi_t) + \beta_{-t} \cos(\langle -\boldsymbol{\omega}_t, \boldsymbol{r} \rangle + \phi_{-t}) \\
&= \sum_{t=1}^{T} \beta_t \sin\left( \langle \boldsymbol{\omega}_t, \boldsymbol{r} \rangle + \phi_t + \frac{\pi}{2} \right) + \beta_{-t} \cos\left( \langle \boldsymbol{\omega}_t, \boldsymbol{r} \rangle - \phi_{-t} \right) \\
&= \sum_{t=1}^{T} R_t \sin\left( \langle \boldsymbol{\omega}_t, \boldsymbol{r} \rangle + \phi_t' \right).
\end{aligned}
$$

The final line follows from the Auxiliary Angle Formula, with:

$$R_t = \sqrt{A_t^2 + B_t^2}, \ \phi_t' = \arctan\left(\frac{B_t}{A_t}\right),$$

$$A_t = \beta_t \cos(\phi_t + \frac{\pi}{2}) + \beta_{-t} \sin(\phi_{-t}), \ B_t = \beta_t \sin(\phi_t + \frac{\pi}{2}) + \beta_{-t} \cos(\phi_{-t}),$$

where we assume $A_t > 0$ for all $t = 1, 2, \ldots, T$, $R_t$ represents the magnitude, and $\phi_t'$ is the adjusted phase angle. For the case where $A_t \leq 0$, we leave the derivation as an exercise for the reader. Let $\boldsymbol{W}_i^{(1)} = [R_t]_{1 \leq t \leq T}^\top$, $\boldsymbol{\phi}_{\mathcal{T}} = [\phi_t']_{1 \leq t \leq T}$, and $b_i^{(1)} = \zeta$. Then, we can conclude that the statement holds for $i = 1, \ldots, F_1$. Second, we would like to show that given $\boldsymbol{W}_i^{(1)} \in \mathbb{R}^{1 \times T}$ and $b_i^{(1)} \in \mathbb{R}$,

$$\boldsymbol{W}_i^{(1)} \sin(\boldsymbol{\Omega}_{\mathcal{T}} \boldsymbol{r} + \boldsymbol{\phi}_{\mathcal{T}}) + b_i^{(1)} = \tilde{z}_i^{(1)} = \sum_{\boldsymbol{s}^{(1)} \in \mathcal{S}^{(1)}} \beta_{\boldsymbol{s}^{(1)}} \cos\left(\langle \boldsymbol{\Omega}_{\mathcal{T}}^\top \boldsymbol{s}^{(1)}, \boldsymbol{r} \rangle + \phi_{\boldsymbol{s}^{(1)}}\right) + \zeta$$

for some $\{\beta_{\boldsymbol{s}^{(1)}}\}$, $\{\phi_{\boldsymbol{s}^{(1)}}\}$ with cardinality $2T$ and $\zeta$. We can re-express the summation as follows:

$$\boldsymbol{W}_i^{(1)} \sin(\boldsymbol{\Omega}_{\mathcal{T}} \boldsymbol{r} + \boldsymbol{\phi}_{\mathcal{T}}) + b_i^{(1)} = \boldsymbol{W}_i^{(1)} \cos(\boldsymbol{\Omega}_{\mathcal{T}} \boldsymbol{r} + \boldsymbol{\phi}_{\mathcal{T}} - \frac{\boldsymbol{\pi}}{\boldsymbol{2}}) + b_i^{(1)}$$

$$= \sum_{t=1}^{T} [\boldsymbol{W}_i^{(1)}]_t \cos(\langle \boldsymbol{\omega}_t, \boldsymbol{r} \rangle + \phi_t - \frac{\pi}{2}) + b_i^{(1)}$$

$$= \sum_{t=1}^{T} ([\boldsymbol{W}_i^{(1)}]_t - \xi) \cos(\langle \boldsymbol{\omega}_t, \boldsymbol{r} \rangle + \phi_t - \frac{\pi}{2}) + \sum_{t=1}^{T} \xi \cos(\langle -\boldsymbol{\omega}_t, \boldsymbol{r} \rangle - \phi_t + \frac{\pi}{2}) + b_i^{(1)}$$

for $\forall \xi \in \mathbb{R}$. Let $\zeta = b_i^{(1)}$, $\{\beta_{\boldsymbol{s}^{(1)}} \in \mathbb{R} \mid \boldsymbol{s}^{(1)} \in \mathcal{S}^{(1)}\} = \{\xi, \cdots, \xi, [\boldsymbol{W}_i^{(1)}]_1 - \xi, \cdots, [\boldsymbol{W}_i^{(1)}]_T - \xi\}$ and $\{\phi_{\boldsymbol{s}^{(1)}} \in \mathbb{R} \mid \boldsymbol{s}^{(1)} \in \mathcal{S}^{(1)}\} = \{-\phi_1 + \frac{\pi}{2}, \cdots, -\phi_T + \frac{\pi}{2}, \phi_1 - \frac{\pi}{2}, \cdots, \phi_T - \frac{\pi}{2}\}$ be two ordered sets. Then, we can conclude that the statement holds for $i = 1, \ldots, F_1$.

Next, we will prove Theorem 4.1 by induction using the previous statement.

**Base case** Let us denote the pre-activation vector at layer $\ell$ as $\tilde{\boldsymbol{z}}^{(\ell)}$, i.e., $\boldsymbol{z}^{(\ell)} = \rho^{(\ell)}(\tilde{\boldsymbol{z}}^{(\ell)})$. Consider the pre-activation of a node at the first layer of the neural network for any mapping of the form in (3). Then

$$\tilde{z}_i^{(1)} = \boldsymbol{W}_i^{(1)} \gamma(\boldsymbol{r}) = \sum_{t=1}^{T} w_{it} \cos\left(\langle \boldsymbol{\omega}_t, \boldsymbol{r} \rangle + \phi_t\right),$$

with some $w_{it} \in \mathbb{R}$ depending on the first layer weights connected to that node and $\phi_t \in \mathbb{R}$. Also note that interchanging sines with cosines only affects the phase terms. After applying the activation function, and using the previous statement and the result of Lemma A.3, the output of each node at the first layer is given by

$$z_i^{(1)} = \rho^{(1)}\left(\tilde{z}_i^{(1)}\right) = \sum_{q=0}^{Q} \alpha_q \left(\tilde{z}_i^{(1)}\right)^q = \sum_{q=0}^{Q} \alpha_q \left(\sum_{t=1}^{T} w_{it} \cos\left(\langle \boldsymbol{\omega}_t, \boldsymbol{r} \rangle + \phi_t\right)\right)^q$$

$$= \sum_{\bar{\boldsymbol{s}}^{(Q)} \in \bar{\mathcal{S}}^{(Q)}} \tilde{\beta}_{\bar{\boldsymbol{s}}^{(Q)}} \cos\left(\langle \boldsymbol{\Omega}_{\mathcal{T}}^\top \bar{\boldsymbol{s}}^{(Q)}, \boldsymbol{r} \rangle + \tilde{\phi}_{\bar{\boldsymbol{s}}^{(Q)}}\right),$$

where $\bar{\mathcal{S}}^{(Q)} = \bigcup_{q=1}^{Q} \mathcal{S}^{(q)} = \mathcal{U}^{(Q)}$ is defined in Lemma A.3. Therefore, the statement trivially holds, i.e.,

$$\bar{\mathcal{S}}^{(Q)} = \left\{ \left[\bar{s}_1^{(Q)}, \cdots, \bar{s}_T^{(Q)}\right]^\top \ \middle| \ \bar{s}_t^{(Q)} \in \mathbb{Z} \wedge \sum_{t=1}^{T} |\bar{s}_t^{(Q)}| \leq Q \right\}.$$

**Induction step** Assume the output of the nodes at layer $\ell$ satisfy the following expression:

$$z_i^{(\ell)} = \sum_{\bar{\boldsymbol{s}}^{(Q^\ell)} \in \bar{\mathcal{S}}^{(Q^\ell)}} \tilde{\beta}_{\bar{\boldsymbol{s}}^{(Q^\ell)}, i} \cos\left(\langle \boldsymbol{\Omega}_{\mathcal{T}}^\top \bar{\boldsymbol{s}}^{(Q^\ell)}, \boldsymbol{r} \rangle + \tilde{\phi}_{\bar{\boldsymbol{s}}^{(Q^\ell)}}\right),$$

where

$$\bar{\mathcal{S}}^{(Q^\ell)} = \left\{ \left[ \bar{s}_1^{(Q^\ell)}, \cdots, \bar{s}_T^{(Q^\ell)} \right]^\top \ \middle| \ \bar{s}_t^{(Q^\ell)} \in \mathbb{Z} \wedge \sum_{t=1}^{T} |\bar{s}_t^{(Q^\ell)}| \leq Q^\ell \right\}.$$

Then, the pre-activation of any node at the $(\ell+1)^{\text{th}}$ layer can be expressed as:

$$\tilde{z}_i^{(\ell+1)} = \sum_{\bar{s}^{(Q^\ell)} \in \bar{\mathcal{S}}^{(Q^\ell)}} \tilde{\bar{\beta}}_{\bar{s}^{(Q^\ell)},i} \cos\left( \langle \boldsymbol{\Omega}_\mathcal{T}^\top \bar{s}^{(Q^\ell)}, r \rangle + \tilde{\bar{\phi}}_{\bar{s}^{(Q^\ell)}} \right),$$

since the sum of sines/cosines with the same frequency only result in a sine/cosine with the same frequency but with a modified phase and amplitude. Hence, after applying the activation function, the output of the $i^{\text{th}}$ node at the $(\ell+1)^{\text{th}}$ layer can be written as:

$$z_i^{(\ell+1)} = \rho^{(\ell+1)}\left( \tilde{z}_i^{(\ell+1)} \right) = \sum_{q=0}^{Q} \alpha_q \left( \sum_{\bar{s}^{(Q^\ell)} \in \bar{\mathcal{S}}^{(Q^\ell)}} \tilde{\bar{\beta}}_{\bar{s}^{(Q^\ell)},i} \cos\left( \langle \boldsymbol{\Omega}_\mathcal{T}^\top \bar{s}^{(Q^\ell)}, r \rangle + \tilde{\bar{\phi}}_{\bar{s}^{(Q^\ell)}} \right) \right)^q.$$

By using Lemma A.4, we have:

$$\left( \sum_{\bar{s}^{(Q^\ell)} \in \bar{\mathcal{S}}^{(Q^\ell)}} \tilde{\bar{\beta}}_{\bar{s}^{(Q^\ell)},i} \cos\left( \langle \boldsymbol{\Omega}_\mathcal{T}^\top \bar{s}^{(Q^\ell)}, r \rangle + \tilde{\bar{\phi}}_{\bar{s}^{(Q^\ell)}} \right) \right)^q = \sum_{\bar{s}^{(qQ^\ell)} \in \bar{\mathcal{S}}^{(qQ^\ell)}} \beta'_{\bar{s}^{(qQ^\ell)},i} \cos\left( \langle \boldsymbol{\Omega}_\mathcal{T}^\top \bar{s}^{(qQ^\ell)}, r \rangle + \phi'_{\bar{s}^{(qQ^\ell)}} \right),$$

where $\bar{\mathcal{S}}^{(qQ^\ell)} = \left\{ \left[ \bar{s}_1^{(qQ^\ell)}, \cdots, \bar{s}_T^{(qQ^\ell)} \right]^\top \ \middle| \ \bar{s}_t^{(qQ^\ell)} \in \mathbb{Z} \wedge \sum_{t=1}^{T} |\bar{s}_t^{(qQ^\ell)}| \leq qQ^\ell \right\}.$

Now, let us use the above result to complete the proof of the inductive step. In particular, we can now express $z_i^{(\ell+1)}$ as:

$$z_i^{(\ell+1)} = \sum_{q=0}^{Q} \alpha_q \sum_{\bar{s}^{(qQ^\ell)} \in \bar{\mathcal{S}}^{(qQ^\ell)}} \beta'_{\bar{s}^{(qQ^\ell)},i} \cos\left( \langle \boldsymbol{\Omega}_\mathcal{T}^\top \bar{s}^{(qQ^\ell)}, r \rangle + \phi'_{\bar{s}^{(qQ^\ell)}} \right)$$

$$= \sum_{s' \in \mathcal{S}'} \beta''_{s',i} \sin\left( \langle \boldsymbol{\Omega}_\mathcal{T}^\top s', r \rangle + \phi''_{s',i} \right),$$

where

$$\mathcal{S}' := \bigcup_{q=1}^{Q} \bar{\mathcal{S}}^{(qQ^\ell)} = \bar{\mathcal{S}}^{(QQ^\ell)} = \bar{\mathcal{S}}^{(Q^{\ell+1})} = \left\{ \left[ \bar{s}_1^{(Q^{\ell+1})}, \cdots, \bar{s}_T^{(Q^{\ell+1})} \right]^\top \ \middle| \ \bar{s}_t^{(Q^{\ell+1})} \in \mathbb{Z} \wedge \sum_{t=1}^{T} |\bar{s}_t^{(Q^{\ell+1})}| \leq Q^{\ell+1} \right\}.$$

This sequence of inclusions concludes the proof of induction. Thus, considering $\gamma(r) = \sin(\boldsymbol{\Omega}_\mathcal{T} r + \phi_\mathcal{T})$, the INR architecture of the form (3) can only represent functions of the form:

$$f_\Theta(r) = \sum_{s \in \mathcal{S}_\mathcal{T}} c_s \sin\left( \langle \boldsymbol{\Omega}_\mathcal{T}^\top s, r \rangle + \phi_s \right),$$

where $\mathcal{S}_\mathcal{T} = \left\{ [s_1, s_2, \ldots, s_T]^\top \ \middle| \ s_t \in \mathbb{Z}, \ \sum_{t=1}^{T} |s_t| \leq Q^{L-1} \right\}.$ $\qquad \square$

### A.7. Proof of the connection between the expressive power of INRs and certain period-2 functions.

*Proof.* As we previously mentioned, interchanging sines with cosines only affects the phase term, we can rewrite the positional encoding in (5) as

$$\gamma(r) = \begin{bmatrix} \sin(\boldsymbol{\Omega} r) \\ \cos(\boldsymbol{\Omega} r) \end{bmatrix} = \begin{bmatrix} \sin(\boldsymbol{\Omega} r) \\ \sin(\boldsymbol{\Omega} r + \frac{\pi}{2} \mathbf{1}_{2T \times 1}) \end{bmatrix} = \sin(\tilde{\boldsymbol{\Omega}} r + \phi),$$

where $\tilde{\boldsymbol{\Omega}} = \left[ \boldsymbol{\Omega}^T \ \boldsymbol{\Omega}^T \right]^T \in \mathbb{R}^{4T \times 2}$, and $\boldsymbol{\phi} = \left[ \mathbf{0}_{2T \times 1}^T \ \frac{\pi}{2} \mathbf{1}_{2T \times 1}^T \right]^T \in \mathbb{R}^{4T \times 1}$. We use the same architecture of the form (3). Directly using the result of Theorem 4.1, the expressive power of this architecture is of the form:

$$f_{\boldsymbol{\Theta}}(\boldsymbol{r}) = \sum_{\boldsymbol{s}' \in \mathcal{S}'_{\mathcal{T}}} c_{\boldsymbol{s}'} \sin \left( \langle \tilde{\boldsymbol{\Omega}}^\top \boldsymbol{s}', \boldsymbol{r} \rangle + \phi_{\boldsymbol{s}'} \right),$$

where

$$\mathcal{S}'_{\mathcal{T}} = \left\{ [s'_1, s'_2, \ldots, s'_{4T}]^\top \ \middle| \ s'_t \in \mathbb{Z}, \ \sum_{t=1}^{4T} |s'_t| \leq Q^{L-1} \right\}.$$

Using the Trigonometric Sum and Difference Formulas, we can rewrite the above as:

$$\begin{aligned} f_{\boldsymbol{\Theta}}(\boldsymbol{r}) &= \sum_{\boldsymbol{s}' \in \mathcal{S}'_{\mathcal{T}}} c_{\boldsymbol{s}'} \sin \left( \langle \tilde{\boldsymbol{\Omega}}^\top \boldsymbol{s}', \boldsymbol{r} \rangle + \phi_{\boldsymbol{s}'} \right) \\ &= \sum_{\boldsymbol{s}' \in \mathcal{S}'_{\mathcal{T}}} c_{\boldsymbol{s}'} \cos \left( \phi_{\boldsymbol{s}'} \right) \sin \left( \langle \tilde{\boldsymbol{\Omega}}^\top \boldsymbol{s}', \boldsymbol{r} \rangle \right) + c_{\boldsymbol{s}'} \sin \left( \phi_{\boldsymbol{s}'} \right) \cos \left( \langle \tilde{\boldsymbol{\Omega}}^\top \boldsymbol{s}', \boldsymbol{r} \rangle \right) \\ &= \sum_{\boldsymbol{s}' \in \mathcal{S}'_{\mathcal{T}}} d_{\boldsymbol{s}'} \sin \left( \langle \tilde{\boldsymbol{\Omega}}^\top \boldsymbol{s}', \boldsymbol{r} \rangle \right) + f_{\boldsymbol{s}'} \cos \left( \langle \tilde{\boldsymbol{\Omega}}^\top \boldsymbol{s}', \boldsymbol{r} \rangle \right). \end{aligned}$$

The inner product $\langle \tilde{\boldsymbol{\Omega}}^\top \boldsymbol{s}', \boldsymbol{r} \rangle$, where $\boldsymbol{r} = [x, y]^\top$, can be expressed as a linear combination of the corresponding components, involving coordinate $x$ and $y$ scaled by the respective elements of $\tilde{\boldsymbol{\Omega}}^\top \boldsymbol{s}'$. Then, we have

$$f_{\boldsymbol{\Theta}}(\boldsymbol{r}) = \sum_{|i| + |j| \leq N, i, j \in \mathbb{Z}} D_{i,j} \sin \left( \pi (ix + jy) \right) + F_{i,j} \cos \left( \pi (ix + jy) \right), \tag{19}$$

where $D_{i,j}$, $F_{i,j}$ are some constants with respect to $i, j$, and $N = \mathcal{O}(2^{T-1} Q^{L-1})$. This can be easily verified using the idea of binary representation, since the frequency matrix $\boldsymbol{\Omega}$ only contains coordinate-wise frequencies $2^t \pi$, $t = 0, \ldots, T-1$. Hence, as the layer of MLPs/INRs goes to infinity, i.e. $L \to \infty$, we have $f_{\boldsymbol{\Theta}}(\boldsymbol{r})$ approaching to (7). $\qquad \square$

## B. Proof of Theorem 4.5

### B.1. Notations and Definitions

Consider an array snapshot containing $K$ targets with azimuth angle $\theta_k$ and elevation angle $\phi_k$ ($k = 1, \cdots, K$). Let $[0, U_1] \times [0, U_2]$ represent the antenna array response field with bounded domain positioned in the $x - y$ plane, consider a general case of an uniform sampling grid of dimensions $M_1 \times M_2$, with spacing $d_1 = \frac{U_1}{M_1 - 1} \leq \frac{\lambda}{2}$ and $d_2 = \frac{U_2}{M_2 - 1} \leq \frac{\lambda}{2}$. According to (2), the $(m_1, m_2)$ th element of the response with respect to $K$ targets in absence of noise can be written as

$$y_{m_1, m_2} = \sum_{k=1}^{K} x_k e^{j \frac{2\pi}{\lambda} \left( (m_1 - 1) d_1 \sin \phi_k \cos \theta_k + (m_2 - 1) d_2 \sin \phi_k \sin \theta_k \right)}$$

for $1 \leq m_1 \leq M_1$ and $1 \leq m_2 \leq M_2$. Notably, when $d_1 = d_2 = \frac{\lambda}{2}$, the sampling pattern aligns with the Nyquist sampling. Let $\mathbf{Y} = [y_{m_1, m_2}]_{1 \leq m_1 \leq M_1, 1 \leq m_2 \leq M_2} \in \mathbb{C}^{M_1 \times M_2}$ be the response matrix with entries as the antenna array response defined in (8).

Given $\mathbf{Y} = [y_{m_1, m_2}] \in \mathbb{C}^{M_1 \times M_2}$ for $1 \leq m_1 \leq M_1, 1 \leq m_2 \leq M_2$, a **Block Hankel matrix** of $\mathbf{Y}$ can be constructed as:

$$\mathcal{H}_{N_1, N_2}(\mathbf{Y}) = \begin{bmatrix} \mathcal{H}_{N_2}(\mathbf{y}_1) & \mathcal{H}_{N_2}(\mathbf{y}_2) & \cdots & \mathcal{H}_{N_2}(\mathbf{y}_{M_1 - N_1 + 1}) \\ \mathcal{H}_{N_2}(\mathbf{y}_2) & \mathcal{H}_{N_2}(\mathbf{y}_3) & \cdots & \mathcal{H}_{N_2}(\mathbf{y}_{M_1 - N_1 + 2}) \\ \vdots & \vdots & \ddots & \vdots \\ \mathcal{H}_{N_2}(\mathbf{y}_{N_1}) & \mathcal{H}_{N_2}(\mathbf{y}_{N_1 + 1}) & \cdots & \mathcal{H}_{N_2}(\mathbf{y}_{M_1}) \end{bmatrix},$$

where $\mathcal{H}_{N_2}(\mathbf{y}_m)$ is defined as:

$$\mathcal{H}_{N_2}(\mathbf{y}_m) = \begin{bmatrix} y_{m,1} & y_{m,2} & \cdots & y_{m,M_2-N_2+1} \\ y_{m,2} & y_{m,3} & \cdots & y_{m,M_2-N_2+2} \\ \vdots & \vdots & \ddots & \vdots \\ y_{m,N_2} & y_{m,N_2+1} & \cdots & y_{m,M_2} \end{bmatrix}.$$

A block Hankel matrix can also be constructed along the column direction, which is defined as:

$$\tilde{\mathcal{H}}_{\tilde{N}_1,\tilde{N}_2}(\mathbf{Y}) = \begin{bmatrix} \tilde{\mathcal{H}}_{\tilde{N}_2}(\mathbf{y}^{(1)}) & \tilde{\mathcal{H}}_{\tilde{N}_2}(\mathbf{y}^{(2)}) & \cdots & \tilde{\mathcal{H}}_{\tilde{N}_2}(\mathbf{y}^{(M_2-\tilde{N}_1+1)}) \\ \tilde{\mathcal{H}}_{\tilde{N}_2}(\mathbf{y}^{(2)}) & \tilde{\mathcal{H}}_{\tilde{N}_2}(\mathbf{y}^{(3)}) & \cdots & \tilde{\mathcal{H}}_{\tilde{N}_2}(\mathbf{y}^{(M_2-\tilde{N}_1+2)}) \\ \vdots & \vdots & \ddots & \vdots \\ \tilde{\mathcal{H}}_{\tilde{N}_2}(\mathbf{y}^{(\tilde{N}_1)}) & \tilde{\mathcal{H}}_{\tilde{N}_2}(\mathbf{y}^{(\tilde{N}_1+1)}) & \cdots & \tilde{\mathcal{H}}_{\tilde{N}_2}(\mathbf{y}^{(M_2)}) \end{bmatrix},$$

where $\tilde{\mathcal{H}}_{\tilde{N}_2}(\mathbf{y}^{(m)})$ is defined as:

$$\tilde{\mathcal{H}}_{\tilde{N}_2}(\mathbf{y}^{(m)}) = \begin{bmatrix} y_{1,m} & y_{2,m} & \cdots & y_{M_1-\tilde{N}_2+1,m} \\ y_{2,m} & y_{3,m} & \cdots & y_{M_1-\tilde{N}_2+2,m} \\ \vdots & \vdots & \ddots & \vdots \\ y_{\tilde{N}_2,m} & y_{\tilde{N}_2+1,m} & \cdots & y_{M_1,m} \end{bmatrix}.$$

Note that $\tilde{H}_{N_2,N_1}(\mathbf{Y}^\top) = \mathcal{H}_{N_1,N_2}(\mathbf{Y})$. Moreover, we have $\text{rank}(\mathcal{H}_{N_1,N_2}(\mathbf{Y})) = \text{rank}(\tilde{\mathcal{H}}_{\tilde{N}_1,\tilde{N}_2}(\mathbf{Y}))$ based on the conditions in Lemma B.1.

For the sake of clarity, we define

$$\boldsymbol{S} = \begin{bmatrix} \boldsymbol{I}_{K\times K} \\ \boldsymbol{0}_K \end{bmatrix} \in \mathbb{R}^{(K+1)\times K}, \, \boldsymbol{b} = \begin{bmatrix} \boldsymbol{0}_K \\ 1 \end{bmatrix} \in \mathbb{R}^{(K+1)\times 1}.$$

The matrices $\boldsymbol{S}$ and $\boldsymbol{b}$ are column selection matrices, selecting the first $K$ columns and the last column, respectively.

## B.2. Lemma B.1, Lemma B.2, Lemma B.3 and their Proofs

**Lemma B.1.** *For the Block Hankel matrix in Definition 4.3, if $K \leq N_1 \leq M_1 - K + 1$ and $K \leq N_2 \leq M_2 - K + 1$, then we have $\text{rank}(\mathcal{H}_{N_1,N_2}(\mathbf{Y})) = K$, and the first $K$ columns of $\mathcal{H}_{N_1,N_2}(\mathbf{Y})$ serve as a basis of $\mathcal{R}(\mathcal{H}_{N_1,N_2}(\mathbf{Y}))$. Similarly, if $K \leq \tilde{N}_1 \leq M_2 - K + 1$ and $K \leq \tilde{N}_2 \leq M_1 - K + 1$, we have $\text{rank}(\tilde{\mathcal{H}}_{\tilde{N}_1,\tilde{N}_2}(\mathbf{Y})) = K$, and the first $K$ columns of $\tilde{\mathcal{H}}_{\tilde{N}_1,\tilde{N}_2}(\mathbf{Y})$ serve as a basis of $\mathcal{R}(\tilde{\mathcal{H}}_{\tilde{N}_1,\tilde{N}_2}(\mathbf{Y}))$.*

*Proof.* Proof followed by (Hua, 1992). $\qquad\square$

**Lemma B.2.** *For Hankel matrix $\mathcal{H}_{N_2}(\mathbf{y}_m)$ $(1 \leq m \leq M_1)$, if $K \leq N_2 \leq M_2 - K + 1$, then $\text{rank}(\mathcal{H}_{N_2}(\mathbf{y}_m)) = K$, and the first $K$ columns of $\mathcal{H}_{N_2}(\mathbf{y}_m)$ serve as a basis of $\mathcal{R}(\mathcal{H}_{N_2}(\mathbf{y}_m))$. Similarly, for Hankel matrix $\tilde{\mathcal{H}}_{\tilde{N}_2}(\mathbf{y}^{(m)})$ $(1 \leq m \leq M_2)$, if $K \leq \tilde{N}_2 \leq M_1 - K + 1$, $\text{rank}(\tilde{\mathcal{H}}_{\tilde{N}_2}(\mathbf{y}^{(m)})) = K$, and the first $K$ columns of $\tilde{\mathcal{H}}_{\tilde{N}_2}(\mathbf{y}^{(m)})$ serve as a basis of $\mathcal{R}(\tilde{\mathcal{H}}_{\tilde{N}_2}(\mathbf{y}^{(m)}))$.*

*Proof.* We will prove the statement for $\mathcal{H}_{N_2}(\mathbf{y}_m)$, while the proof for $\tilde{\mathcal{H}}_{\tilde{N}_2}(\mathbf{y}^{(m)})$ follows the same way therefore omitted here. According to (8), we have

$$\mathbf{y}_m = \begin{bmatrix} 1 & 1 & \cdots & 1 \\ e^{j\frac{2\pi}{\lambda}d_1\sin\phi_1\sin\theta_1} & e^{j\frac{2\pi}{\lambda}d_1\sin\phi_2\sin\theta_2} & \cdots & e^{j\frac{2\pi}{\lambda}d_1\sin\phi_K\sin\theta_K} \\ \vdots & \vdots & \ddots & \vdots \\ e^{j\frac{2\pi}{\lambda}(M_2-1)d_1\sin\phi_1\sin\theta_1} & e^{j\frac{2\pi}{\lambda}(M_2-1)d_1\sin\phi_2\sin\theta_2} & \cdots & e^{j\frac{2\pi}{\lambda}(M_2-1)d_1\sin\phi_K\sin\theta_K} \end{bmatrix} \begin{bmatrix} x_1 e^{j\frac{2\pi}{\lambda}(m-1)d_1\sin\phi_1\cos\theta_1} \\ x_2 e^{j\frac{2\pi}{\lambda}(m-1)d_1\sin\phi_2\cos\theta_2} \\ \vdots \\ x_K e^{j\frac{2\pi}{\lambda}(m-1)d_1\sin\phi_K\cos\theta_K} \end{bmatrix}$$

$$= \mathbf{A}_{M_2}(\boldsymbol{\theta},\boldsymbol{\phi})\mathbf{s}_m.$$

It can be shown that $\mathcal{H}_{N_2}(\mathbf{y}_m)$ admits a Vandermonde decomposition

$$\mathcal{H}_{N_2}(\mathbf{y}_m) = \mathbf{A}_{N_2}(\boldsymbol{\theta}, \boldsymbol{\phi})\mathrm{diag}(\mathbf{s}_m)(\mathbf{A}_{M_2-N_2+1}(\boldsymbol{\theta}, \boldsymbol{\phi}))^T,$$

where $\mathbf{A}_{N_2}(\boldsymbol{\theta}, \boldsymbol{\phi})$ and $\mathbf{A}_{M_2-N_2+1}(\boldsymbol{\theta}, \boldsymbol{\phi})$ are both Vandermonde matrix. It is easy to verify that if $K \leq N_2 \leq M_2 - K + 1$, $\mathrm{rank}(\mathcal{H}_{N_2}(\mathbf{y}_m)) = K$. And the first $K$ columns of $\mathcal{H}_{N_2}(\mathbf{y}_m)$ have the form $\mathbf{A}_{N_2}(\boldsymbol{\theta}, \boldsymbol{\phi})\mathrm{diag}(\mathbf{s}_m)(\mathbf{A}_K(\boldsymbol{\theta}, \boldsymbol{\phi}))^T$, which can be verified to be rank-$K$ due to the Vandermonde structure. Thus, the first $K$ columns form a linearly independent set, therefore serve as a basis for $\mathcal{R}(\mathcal{H}_{N_2}(\mathbf{y}_m))$. $\qquad\square$

**Lemma B.3.** *Consider Hankel matrix $\mathcal{H}_{M_2-K}(\mathbf{y}_m)$ ($1 \leq m \leq M_1$) generated using (8), there exists a unique $\mathbf{m}_1 \in \mathbb{C}^K$ such that*

$$\sum_{m=1}^{M_1} \|\mathcal{H}_{M_2-K}(\mathbf{y}_m)\boldsymbol{S}\mathbf{m}_1 - \mathcal{H}_{M_2-K}(\mathbf{y}_m)\boldsymbol{b}\|_2 = 0. \tag{20}$$

*Similarly, consider Hankel matrix $\tilde{\mathcal{H}}_{M_1-K}(\mathbf{y}^{(m)})$ ($1 \leq m \leq M_2$) generated using (8), there exists a unique $\mathbf{m}_2 \in \mathbb{C}^K$ such that*

$$\sum_{m=1}^{M_2} \|\tilde{\mathcal{H}}_{M_1-K}(\mathbf{y}^{(m)})\boldsymbol{S}\mathbf{m}_2 - \tilde{\mathcal{H}}_{M_1-K}(\mathbf{y}^{(m)})\boldsymbol{b}\|_2 = 0. \tag{21}$$

*Proof.* We will prove the statement for $\mathcal{H}_{M_2-K}(\mathbf{y}_m)$ ($1 \leq m \leq M_1$), while the proof for $\tilde{\mathcal{H}}_{M_1-K}(\mathbf{y}^{(m)})$ follows the same way therefore omitted here. According to Lemma B.2, we have $\mathrm{rank}(\mathcal{H}_{M_2-K}(\mathbf{y}_m)) = K$ where its first $K$ columns serve as a basis of $\mathcal{R}(\mathcal{H}_{M_2-K}(\mathbf{y}_m))$. This means there exists $\boldsymbol{\alpha} = [\alpha_1, \cdots, \alpha_K]^T$ such that $\mathcal{H}_{M_2-K}(\mathbf{y}_m)\boldsymbol{S}\boldsymbol{\alpha} = \mathcal{H}_{M_2-K}(\mathbf{y}_m)\boldsymbol{b}$. Now let us analyze whether $\boldsymbol{\alpha}$ depend on $m$. Using Vandermonde decomposition and explicit form of least squares solution, we have

$$\mathcal{H}_{M_2-K}(\mathbf{y}_m)\boldsymbol{S} = \mathbf{A}_{M_2-K}(\boldsymbol{\theta}, \boldsymbol{\phi})\mathrm{diag}(\mathbf{s}_m)(\mathbf{A}_K(\boldsymbol{\theta}, \boldsymbol{\phi}))^T,$$

$$\mathcal{H}_{M_2-K}(\mathbf{y}_m)\boldsymbol{b} = \mathbf{A}_{M_2-K}(\boldsymbol{\theta}, \boldsymbol{\phi})\mathrm{diag}(\mathbf{s}_m)\begin{bmatrix} e^{j\frac{2\pi}{\lambda}Kd_1\sin\phi_1\sin\theta_1} \\ \vdots \\ e^{j\frac{2\pi}{\lambda}Kd_1\sin\phi_K\sin\theta_K} \end{bmatrix},$$

$$\boldsymbol{\alpha} = \left((\mathcal{H}_{M_2-K}(\mathbf{y}_m)\boldsymbol{S})^H(\mathcal{H}_{M_2-K}(\mathbf{y}_m)\boldsymbol{S})\right)^{-1}(\mathcal{H}_{M_2-K}(\mathbf{y}_m)\boldsymbol{S})^H\mathcal{H}_{M_2-K}(\mathbf{y}_m)\boldsymbol{b}$$

$$= (\mathbf{A}_K^T(\boldsymbol{\theta}, \boldsymbol{\phi}))^{-1}\begin{bmatrix} e^{j\frac{2\pi}{\lambda}Kd_1\sin\phi_1\sin\theta_1} \\ \vdots \\ e^{j\frac{2\pi}{\lambda}Kd_1\sin\phi_K\sin\theta_K} \end{bmatrix}.$$

And we can see that $\boldsymbol{\alpha} \in \mathbb{C}^K$ does not depend on $m$ but only depend on $\boldsymbol{\theta}$ and $\boldsymbol{\phi}$. This means for Hankel matrix $\mathcal{H}_{M_2-K}(\mathbf{y}_m)$ ($1 \leq m \leq M_1$), there exists a unique $\mathbf{m}_1 \in \mathbb{C}^K$ such that

$$\sum_{m=1}^{M_1} \|\mathcal{H}_{M_2-K}(\mathbf{y}_m)\boldsymbol{S}\mathbf{m}_1 - \mathcal{H}_{M_2-K}(\mathbf{y}_m)\boldsymbol{b}\|_2 = 0.$$

$\qquad\square$

### B.3. Main Proof of Theorem 4.5

*Proof.* We will prove the statement for $\mathcal{H}_{N_1,N_2}(\mathbf{Y})$, while the proof for $\tilde{\mathcal{H}}_{\tilde{N}_1,\tilde{N}_2}(\mathbf{Y})$ follows the same way therefore omitted here. Let $N_1 = M_1 - K + 1$ and $N_2 = M_2 - K$, consider the matrix $\mathcal{H}_{M_1-K+1,M_2-K}(\mathbf{Y})$. According to Lemma B.1, we have $\mathrm{rank}(\mathcal{H}_{M_1-K+1,M_2-K}(\mathbf{Y})) = K$, and its first $K$ columns

$$\mathcal{H}_{M_1-K+1,M_2-K}(\mathbf{Y})\boldsymbol{S} = \begin{bmatrix} \mathcal{H}_{M_2-K}(\mathbf{y}_1)\boldsymbol{S} \\ \mathcal{H}_{M_2-K}(\mathbf{y}_2)\boldsymbol{S} \\ \vdots \\ \mathcal{H}_{M_2-K}(\mathbf{y}_{M_1-K+1})\boldsymbol{S} \end{bmatrix}$$

are linear independent. If we keep appending columns at the bottom of $\mathcal{H}_{M_1-K+1,M_2-K}(\mathbf{Y})\mathbf{S}$, the columns of the resulting matrix

$$\mathcal{H}_{M_1,M_2-K}(\mathbf{Y})\mathbf{S} = \begin{bmatrix} \mathcal{H}_{M_2-K}(\mathbf{y}_1)\mathbf{S} \\ \vdots \\ \mathcal{H}_{M_2-K}(\mathbf{y}_{M_1-K+1})\mathbf{S} \\ \mathcal{H}_{M_2-K}(\mathbf{y}_{M_1-K+2})\mathbf{S} \\ \vdots \\ \mathcal{H}_{M_2-K}(\mathbf{y}_{M_1})\mathbf{S} \end{bmatrix}$$

is still linear independent. Using Lemma B.2 and Lemma B.3, if we add the $K+1$-th column at the right of the above matrix, the resulting matrix $\mathcal{H}_{M_1,M_2-K}(\mathbf{Y})$ is still rank-$K$ with its first $K$ columns serve as a basis of $\mathcal{R}(\mathcal{H}_{M_1,M_2-K}(\mathbf{Y}))$. Thus, there exists a unique global optimizer $\mathbf{m}_1^o$ such that

$$\|\mathcal{H}_{M_1,M_2-K}(\mathbf{Y})\mathbf{S}\mathbf{m}_1^o - \mathcal{H}_{M_1,M_2-K}(\mathbf{Y})\mathbf{b}\|_2 = 0.$$

$\square$

# C. Further Experimental Results and Details

## C.1. Experimental Setup

### C.1.1. SIMULATION DATA GENERATION

For the signal model in (2), we assume the reflection coefficients follow a circularly symmetric complex Gaussian distribution, given by $x_k \sim \mathcal{CN}(0,\sigma_x^2)$ with $\sigma_x = 1$ for $k = 1,\ldots,K$. The additive noise is modeled as $n_{i,j} \sim \mathcal{CN}(0,\sigma_n^2)$, where $\sigma_n$ is determined by the specified SNR levels. The SNR is defined as:

$$\text{SNR}_{\text{dB}} = 10\log_{10}\frac{P_x}{P_n} = 10\log_{10}\frac{\sigma_x^2}{\sigma_n^2}.$$

For tasks with different sampling configurations, we use the following selected indices for sub-sampling:

- $6 \times 6$: $\mathcal{S}_x = \mathcal{S}_y = \{0,1,2,3,11,19\}$,

- $8 \times 8$: $\mathcal{S}_x = \mathcal{S}_y = \{0,1,2,3,4,9,14,19\}$,

- $10 \times 10$: $\mathcal{S}_x = \mathcal{S}_y = \{0,1,2,3,4,7,10,13,16,19\}$.

**Remark.** When the index sets are mapped to the world coordinate system, each discrete index $(i,j)$ corresponds to the physical position

$$\mathbf{r}_{ij} = \left(i\tfrac{\lambda}{2}, j\tfrac{\lambda}{2}\right),$$

where $\lambda$ is the wavelength.

For angle resolution experiments, we define the azimuth and elevation angles of two targets as $[10,20]$ and $[10+\Delta, 20+\Delta]$ degrees, where $\Delta$ represents the angular separation, which varies from 3 to 10 degrees. For the other tasks, the azimuth and elevation angles of each target are randomly sampled from a uniform distribution over $[-60, 60]$ degrees. Each experiment is conducted with $N = 50$ Monte-Carlo trails.

### C.1.2. EVALUATION METRICS

The Normalized Root Mean Square Error (NRMSE) is defined as:

$$\text{NRMSE} = \frac{1}{N}\sum_{n=1}^{N}\frac{\|\hat{\mathbf{Y}}_n - \mathbf{Y}_n\|_F}{\|\mathbf{Y}_n\|_F},$$

where $\hat{Y}$ and $Y$ denote the predicted array response and the ground truth full *virtual* array, respectively, and $\|\cdot\|_F$ represents the Frobenius norm.

The Resolution Probability is defined as:

$$\text{RP} = \frac{1}{N} \sum_{n=1}^{N} \mathbf{1}_{\{\mathbb{E}_a \wedge \mathbb{E}_e\}}, \text{with}$$

$$\mathbb{E}_a = \left\{ \hat{\theta}_{a,1}^n \in \left[ \theta_{a,1} - \frac{\Delta}{2}, \theta_{a,1} + \frac{\Delta}{2} \right], \hat{\theta}_{a,2}^n \in \left[ \theta_{a,2} - \frac{\Delta}{2}, \theta_{a,2} + \frac{\Delta}{2} \right] \right\},$$

$$\mathbb{E}_e = \left\{ \hat{\theta}_{e,1}^n \in \left[ \theta_{e,1} - \frac{\Delta}{2}, \theta_{e,1} + \frac{\Delta}{2} \right], \hat{\theta}_{e,2}^n \in \left[ \theta_{e,2} - \frac{\Delta}{2}, \theta_{e,2} + \frac{\Delta}{2} \right] \right\},$$

where $\hat{\boldsymbol{\theta}}_i^n = [\hat{\theta}_{a,i}^n, \hat{\theta}_{e,i}^n], i = 1, 2$ denote the prediction azimuth and elevation angles for each target in $n$-th Monte-Carlo trail, $\boldsymbol{\theta}_i = [\theta_{a,i}, \theta_{e,i}], i = 1, 2$ represent the ground truth azimuth and elevation angles, $\mathbf{1}_{\{\cdot\}}$ is the indicator function, and separation $\Delta \overset{\triangle}{=} \theta_{a,2} - \theta_{a,1} = \theta_{e,2} - \theta_{e,1}$.

### C.1.3. Optimization and Hyperparameters

We optimize the loss function defined in (11) through a two-stage training process. In the initial warm-up stage, we set $\lambda = 0$ and optimize using the Adam optimizer with $\beta = (0.9, 0.999)$ and a weight decay of $10^{-4}$. Letting $\boldsymbol{\Theta}_0 = \arg\min_{\boldsymbol{\Theta}} \mathcal{L}_d$, we use the obtained parameters as the initialization for the next stage. In the adaptation/training stage, we optimize $\boldsymbol{\Theta}$, $\boldsymbol{m}_1$, and $\boldsymbol{m}_2$ using Adam with the same configuration as in the warm-up stage. In both the simulation and real-world experiments, we normalized the input coordinates to the range $(-1, 1]$.

We provide detail hyperparameter settings for both simulation and real-world experiments. The model architecture remains consistent with that described in the Experiments section. For simulation tasks, we use a learning rate of $10^{-4}$ and train for $5,000$ epochs in the warm-up stage. In the adaptation stage, we set $\lambda = 0.5$, $\text{lr}_{\boldsymbol{\Theta}} = 10^{-3}$, $\text{lr}_{\boldsymbol{m}_1, \boldsymbol{m}_2} = 3 \times 10^{-3}$, and train for $25,000$ epochs, with K_max set to the exact number of targets for each scenario.

For real-world experiments, we adopt a learning rate of $10^{-4}$ and train for $10,000$ epochs in the warm-up stage. In the adaptation stage, we set $\lambda = 1$, $\text{lr}_{\boldsymbol{\Theta}} = 10^{-3}$, $\text{lr}_{\boldsymbol{m}_1, \boldsymbol{m}_2} = 3 \times 10^{-3}$, and train for $50,000$ epochs. Here, we set K_max $= 4$ as an upper bound on the number of targets in each range bin, as typically, the number of targets within a single range bin is very small (Sun et al., 2020).

### C.2. Implementation Details and Analysis of Baseline Methods

**EMaC.** We adopt equation (9) from the original paper (Chen & Chi, 2013) as the optimization problem, which can be solved using CVX (CVX Research, 2012; Grant & Boyd, 2008) toolbox.

**SIREN.** We adopt the same architecture and recommended hyperparameters from the original paper (Sitzmann et al., 2020). For a fair comparison, we match NEAR's network size, using a depth of $L = 4$ and a hidden layer width of 256.

**NeRF$^2$.** We adopt the same architecture and recommended hyperparameters from the original paper (Zhao et al., 2023). To match our experimental setup, the location of TX is fixed (co-located with RX), and the unknown antenna response is inferred based on their spatial coordinates. The loss function is calculated as the mean-squared error (MSE) between predicted array responses and observed array response, rather than using RSSI values.

**Remark.** The inferior performance of NeRF$^2$ is attributed to some important distinctions between radiance-field reconstruction and our method, which are listed below:

- Our setting uses far fewer measurements (see below) in the form of a antenna array response, compared to NeRF$^2$. This renders measurement-heavy methods like NeRF$^2$ somewhat inferior in our settings. Hence we need to heavily utilize the underlying wave propagation model and the harmonic structure of measurements received at antenna arrays, in order to successfully regularize the problem with so few measurements. This is a major contribution of our work which sets us apart from direct use of NeRF$^2$.

- In fact, NEAR targets a different objective than NeRF$^2$. Our approach emphasizes more on the (super-resolution) localization of the targets, while NeRF$^2$ cares more about the physical property of all objects in a 3D scene in order to model signal propagation. This also serves a crucial reason why we opt to directly predict the response from the antenna coordinates rather than modeling all the voxels' properties as a continuous volumetric function.

- As explained earlier, NeRF$^2$ requires a large set of measurements for training. According to (Zhao et al., 2023), it uses around $6000 \times 21$ measurements and $80\%/20\%$ for training/testing splitting, while we only use a sparse set of $8 \times 8$ measurements for training. Under the same setting of training, NEAR uses less than half of the training time of NeRF$^2$ due to our proposed regularization rather than the ray tracing strategy, which is well known for its heavy computational cost.

### C.3. Additional Details on Radar Data Processing

To sense the environment, the system emits a sequence of waveforms, commonly referred to as chirp signals, through the Tx within a short time interval. These signals propagate, interact with objects in the environment, and are subsequently reflected back to be captured by the Rx. The received signals are then processed to generate an intermediate frequency (IF) signal by mixing the transmitted and received signals from each Tx-Rx antenna pair. This mixed signal is then sampled by an ADC to generate discrete samples for each chirp. By aggregating ADC samples across all chirps and Tx-Rx antenna pairs, the sensing system constructs a three-dimensional (3D) complex data cube for each frame. This data cube is organized into three dimensions: fast time, slow time, and channel, which correspond to range, range rate, and angle, respectively (Kramer et al., 2022).

To process the acquired ADC samples, fourier techniques are applied along the fast time and slow time dimensions to extract detailed information. The first range processing is performed across the fast time axis to isolate objects at different distances into distinct frequency responses within range bins defined by hardware specifications. Subsequently, a Doppler processing along the slow time axis decodes phase variances—Doppler bins—to derive relative radial velocities, producing a range-Doppler (RD) map (Ding et al., 2024). An additional CFAR target detector is usually employed to detect peaks that stand out prominently from their surroundings in the range-Doppler velocity heat-map by comparing local signal power to an adaptive threshold. DOA processing is then performed only for the peaks detected by the CFAR detector.

### C.4. More Experimental Results

Some more experimental results of target localization are shown below.

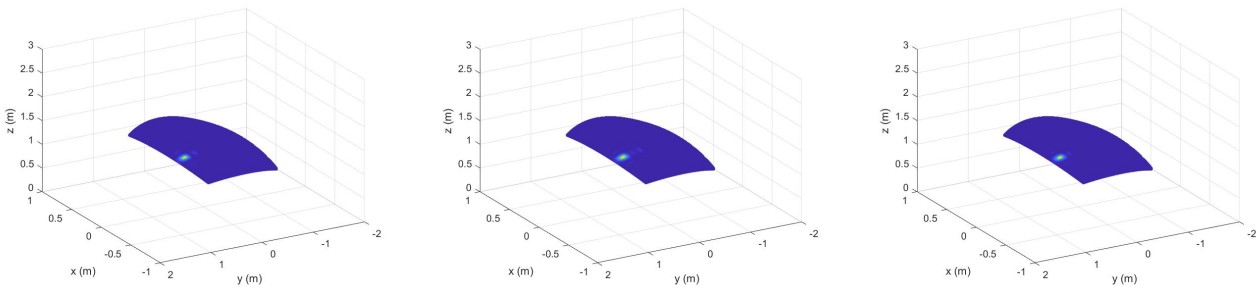

*Figure 6.* Point cloud visualizations for target localization with $K = 1$ (scenario 1). Left: Full array, Middle: NEAR, Right: EMaC.

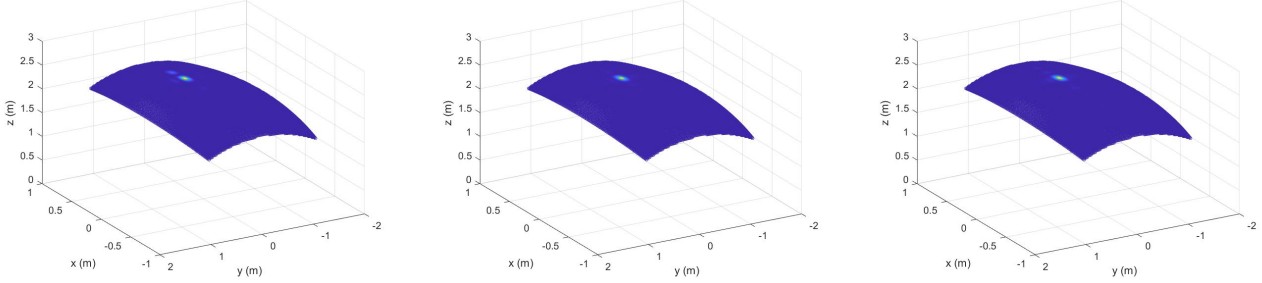

*Figure 7.* Point cloud visualizations for target localization with $K = 1$ (scenario 2). Left: Full array, Middle: NEAR, Right: EMaC.

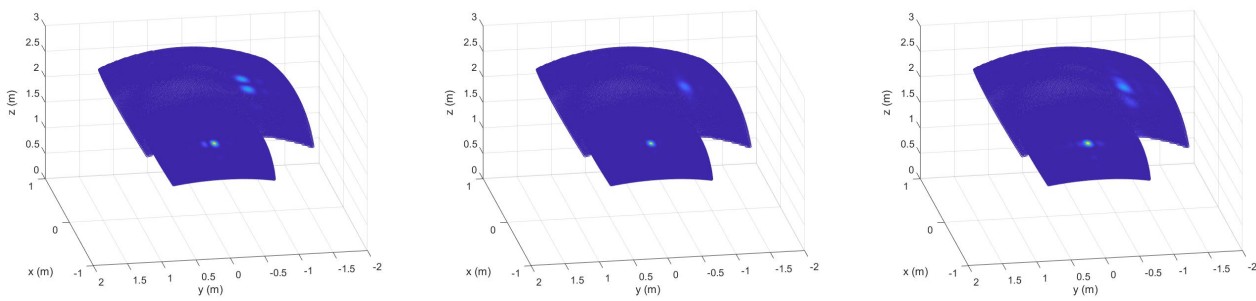

*Figure 8.* Point cloud visualizations for target localization with $K = 2$ (scenario 1). Left: Full array, Middle: NEAR, Right: EMaC.

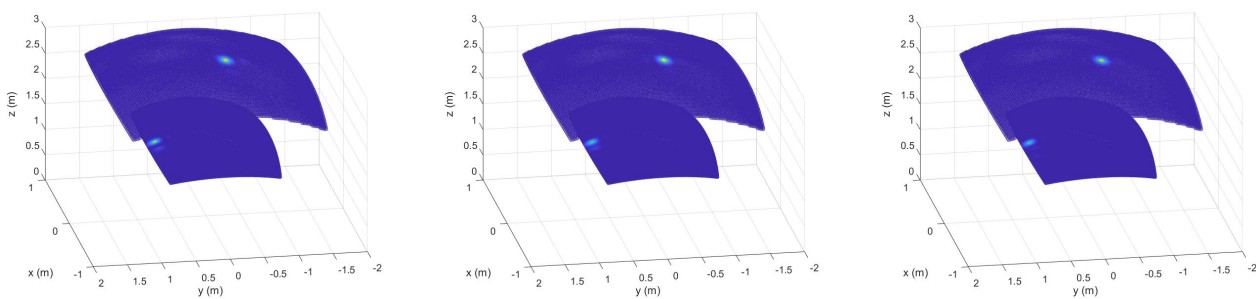

*Figure 9.* Point cloud visualizations for target localization with $K = 2$ (scenario 2). Left: Full array, Middle: NEAR, Right: EMaC.

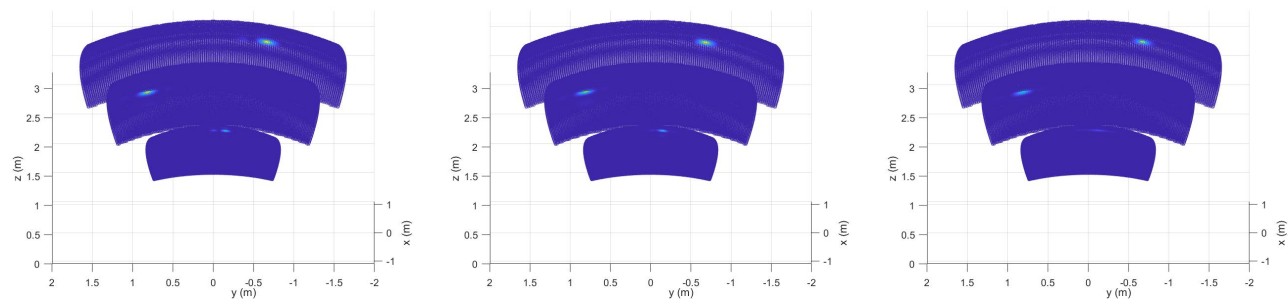

*Figure 10.* Point cloud visualizations for target localization with $K = 3$ (scenario 1). Left: Full array, Middle: NEAR, Right: EMaC.

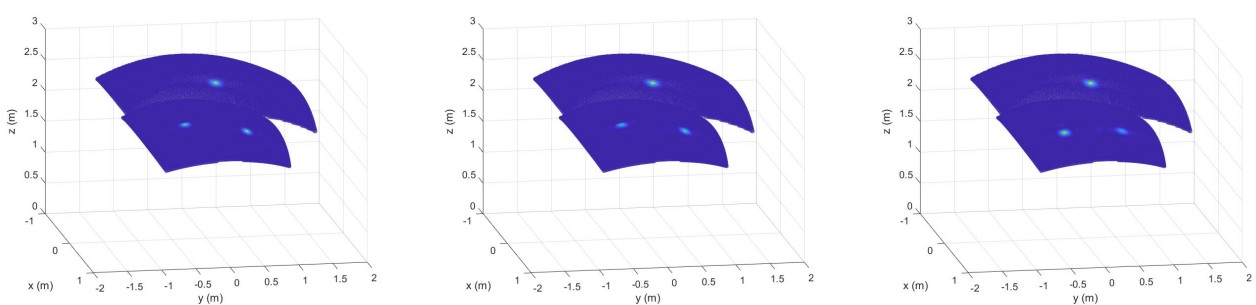

*Figure 11.* Point cloud visualizations for target localization with $K = 3$ (scenario 2). Left: Full array, Middle: NEAR, Right: EMaC.

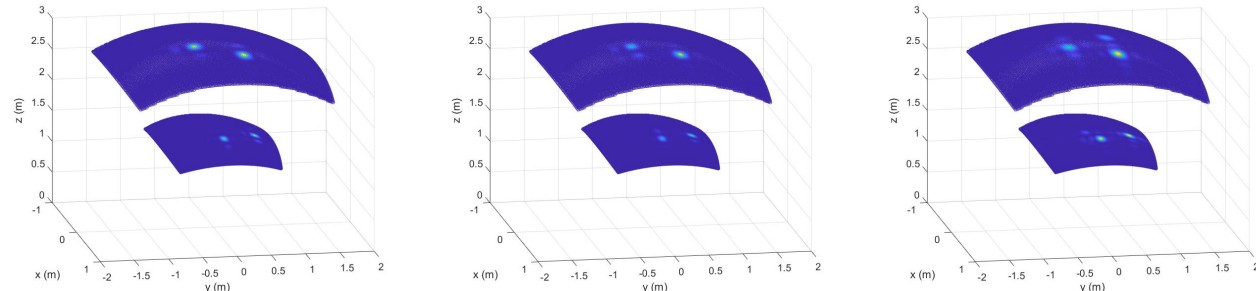

*Figure 12.* Point cloud visualizations for target localization with $K = 4$ (scenario 1). Left: Full array, Middle: NEAR, Right: EMaC.

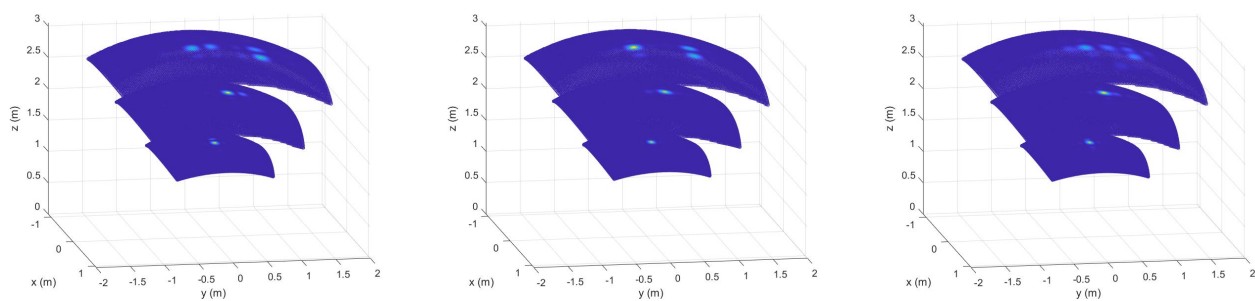

*Figure 13.* Point cloud visualizations for target localization with $K = 4$ (scenario 2). Left: Full array, Middle: NEAR, Right: EMaC.

