# OpenReview forum: "NEAR: Neural Electromagnetic Array Response"
_ICML.cc/2025/Conference — ICML 2025 poster_

### Official Review · Reviewer_moMV · 2025-02-26

**Overall Recommendation:** 3

**Summary:**

Multi-antenna radar systems face challenges in achieving high angular resolution due to hardware constraints, noise, and limited physical antennas. Traditional supervised learning methods for super-resolution struggle with generalization in unseen environments and require extensive training data.

The authors propose Neural Electromagnetic Array Response (NEAR), an untrained implicit neural representation (INR) framework that predicts complex radar responses at arbitrary 2D spatial coordinates using sparse antenna measurements. NEAR predicts radar responses at unobserved locations by exploiting latent harmonic structures in radar wave propagation. It integrates physics-based signal processing with neural networks, avoiding reliance on large datasets.

They claim that this is the first time to establish a link between antenna array response and the expressive power of an INR architecture. Additionally, they introduces a novel regularizer that incorporates radar physics and latent geometry to bridge the gap between traditional INR and NEAR. They include a 20 ×20 full virtual array response (noisy) as a benchmark reference. Evaluated through simulations and real-world (a commercial MIMO radar platform IMAGEVK-74) radar experiments, NEAR demonstrates superior performance in unseen environments compared to conventional methods.

Overall, this research bridges signal processing and INR, offering a data-efficient, physics-aware solution for radar super-resolution without compromising interpretability.

**Claims And Evidence:**

The authors proves that predicting complex-valued responses at any arbitrary location within the 2D virtual antenna array domain is indeed a problem that falls within the class of functions representable by INR.
Furthermore, they show that this mapping can be effectively approximated using multiple layers of Multi-Layer Perceptrons.

To further justify the effectiveness of the physics-informed regularizer, the authors establish
the algebraic properties of the block Hankel matrix corresponding to the ground truth response,
leveraging its harmonic structure.

The experiments demonstrate that the proposed method outperforms the baseline in terms of generalization,
exhibits robustness to noise, achieves superior performance in multi-objective Direction-of-Arrival (DOA) estimation,
and significantly reduces hardware requirements.

**Essential References Not Discussed:**

N/A

**Experimental Designs Or Analyses:**

The validity of the experimental designs can be referenced in the section "Methods and Evaluation Criteria."

It is suggested to include a comparison with a NeRF-like method,
where the rendering operation of NeRF can be modified to model radar waves.
By adapting the NeRF approach for radar response,
it could serve as a valuable baseline for evaluating the proposed method.

**Methods And Evaluation Criteria:**

The proposed method does not rely on large-scale data, exhibits low computational consumption, and demonstrates greater generalizability.

They include a 20 $\times$ 20 full virtual array response (noisy) as a benchmark reference.
A series of validations are conducted in both simulated and real-world environments, encompassing scenarios with varying noise intensities, multiple targets, and different sampling densities.

In conclusion, the authors demonstrate the advantages of the proposed method in addressing the radar response super-resolution problem through a series of well-conducted experiments.

**Other Comments Or Suggestions:**

N/A

**Other Strengths And Weaknesses:**

*Strengths*:

Their work refines the INR superset analysis \[1\] by providing the exact set,
rather than a superset, of integer harmonic frequencies that characterize INR functions.
This derivation delivers a precise and tight characterization of the expressive power of INRs.

They assert that their method employs a straightforward yet effective regularization strategy, in contrast to the more complex ray tracing approach\[2\].


\[1\] Roddenberry, T. M., Saragadam, V., de Hoop, M. V.,
and Baraniuk, R. G. Implicit neural representations
and the algebra of complex wavelets. arXiv preprint
arXiv:2310.00545, 2023.

\[2\] Chen, X., Feng, Z., Sun, K., Qian, K., and Zhang, X. Rf-
Canvas: Modeling RF channel by fusing visual priors
and few-shot RF measurements. In Proceedings of the
22nd ACM Conference on Embedded Networked Sensor
Systems, 2024.

*Weaknesses*:

In this task, both the radar and environmental settings (similar to light and camera parameters in NeRF) are kept constant,
with the goal of predicting higher-resolution radar response maps.
However, the authors do not provide a rationale for why they choose to directly predict the value from the coordinates,
rather than modeling it as a radiance field-like function.
Lack of experiments comparing the two.

**Questions For Authors:**

Why opt to directly predict the value from the coordinates, instead of modeling it as a radiance field-like function?
Please refer to "Experimental Designs Or Analyses".
The superiority of the proposed method would be more effectively highlighted through an experimental comparison with radiance field-like method.

**Relation To Broader Scientific Literature:**

The approach of combining implicit fields with physical constraints is a key feature in many recent works on 3D reconstruction.
For example, the incorporation of continuous medium dynamics to describe the evolution of the Gaussian distribution illustrates how physical principles can be leveraged to enhance the accuracy and realism of reconstructed models\[1\].

\[1\] Tianyi Xie, Zeshun Zong, Yuxin Qiu, Xuan Li, Yutao Feng, Yin Yang, and Chenfanfu Jiang. Physgaussian: Physics-integrated 3d gaussians for generative dynamics. In Proceedings of the IEEE/CVF Conference on Computer Vision and Pattern Recognition (CVPR), 2024.

**Theoretical Claims:**

Theorem 4.1 gives an exact characterization of the set $S_T$ of all possible integer harmonics of the feature mapping $γ(r)$.
The mapping from 2D coordinates to the complex values of the radar response is shown to belong to the class of INRs, as supported by Remark 4.2 and Theorem 4.1.

---

> ### Author Rebuttal · Authors · 2025-04-01
>
> We are grateful for your recognition of our work’s strengths, notably our theoretical analysis that precisely characterizes the expressive power of INR, as well as our development of an efficient and effective regularization strategy. In response to your concern regarding the rationale behind directly predicting the response value from the coordinates—as opposed to modeling it as a radiance field-like function—we have conducted a comprehensive comparison between our approach (NEAR) and NeRF$^2$[1], an innovative extension of NeRF[2] into the electromagnetism domain using ray tracing. NeRF$^2$ constructs a continuous volumetric scene function that interprets the propagation of RF signals and can tell what signal is received at any position after training with a set of input signal measurements.
>
> We compare NeRF$^2$ and our method (NEAR) in terms of angular resolution, target localization accuracy (same as we did in Section 5.2) and average running time using the real-world collected data. We adopt hyperparameters recommended in [1], and run all experiments on a laptop with CPU AMD Ryzen 9 5900 HS with Radeon Graphics and GPU NVIDIA GeForce RTX 3050 Ti Laptop.
> | Method  | Angular resolution for 2m/3m/4.5m | Localization error for 1/2/3/4 target(s) | Average running time |
> |:-------:|:--------------------------------:|:--------------------------------------:|:--------------------:|
> | NEAR    | 5.7248$^\circ$/6.6769$^\circ$/6.9941$^\circ$ | 0.0744m/0.0770m/0.0762m/0.0718m | 550.83s |
> | NeRF$^2$ | 8.5783$^\circ$/8.5783$^\circ$/8.8948$^\circ$ | 0.4902m/0.5096m/0.4346m/0.3898m | 1278.31s |
>
> From the table, we can see that NEAR is able to resolve smaller angle separation at different distances (and signal to noise ratio, SNR) and achieve a much smaller target localization error, compared to NeRF$^2$. This is attributed to some important distinctions between radiance-field reconstruction and our method, which are listed below:
>
> - Our setting uses far fewer measurements (see below) in the form of a antenna array response, compared to NeRF$^2$. This renders measurement-heavy methods like NeRF$^2$ somewhat inferior in our settings. Hence we need to heavily utilize the underlying wave propagation model and the harmonic structure of measurements received at antenna arrays, in order to successfully regularize the problem with so few measurements. This is a major contribution of our work which sets us apart from direct use of NeRF$^2$.
>
> - In fact, NEAR targets a different objective than NeRF$^2$. Our approach emphasizes more on the (super-resolution) localization of the targets, while NeRF$^2$ cares more about the physical property of all objects in a 3D scene in order to model signal propagation. This also serves a crucial reason why we opt to directly predict the response from the antenna coordinates rather than modeling all the voxels' properties as a continuous volumetric function.
>
> - As explained earlier, NeRF$^2$ requires a large set of measurements for training. According to [1], it uses around $6000 \times 21$ measurements and 80\%/20\% for training/testing splitting, while we only use a sparse set of $8\times8$ measurements for training. Under the same setting of training, NEAR uses less than half of the training time of NeRF$^2$ due to our proposed regularization rather than the ray tracing strategy, which is well known for its heavy computational cost.
>
> In summary, NEAR outperforms NeRF$^2$ by directly predicting antenna responses while leveraging the harmonic signal structure. Our approach significantly reduces the required training measurements, leading to improved angular resolution, enhanced localization accuracy, and reduced runtime.
>
> **Reference:**
>
> [1] Zhao, X., An, Z., Pan, Q., \& Yang, L. (2023, October). Nerf2: Neural radio-frequency radiance fields. In Proceedings of the 29th Annual International Conference on Mobile Computing and Networking (pp. 1-15).
>
> [2] Mildenhall, B., Srinivasan, P. P., Tancik, M., Barron, J. T., Ramamoorthi, R., \& Ng, R. (2021). Nerf: Representing scenes as neural radiance fields for view synthesis. Communications of the ACM, 65(1), 99-106.

---

> > ### Comment · Reviewer_moMV · 2025-04-02
> >
> > The authors' response has addressed my proposed concerns. However, as I am not an expert in this field, I tend to keep my original rating (Borderline).

---

### Official Review · Reviewer_xAKz · 2025-03-11

**Overall Recommendation:** 3

**Summary:**

This paper addresses the challenge of achieving high-resolution angular estimation in multi-antenna radar systems using sparse measurements. The authors propose NEAR (Neural Electromagnetic Array Response), an innovative framework that leverages implicit neural representations (INRs) to predict complete antenna array responses from limited physical antenna data, effectively creating a large virtual sensing system with few physical antennas.

The core technical contribution lies in seamlessly integrating INRs with a physics-informed regularization strategy, specifically designing a novel Block Hankel matrix-based constraint that captures the inherent harmonic structures of radar wave propagation. By developing a theoretically grounded approach that maps spatial coordinates to complex-valued antenna responses, the authors enable a continuous representation of the antenna array response field, which allows for super-resolution angular estimation while maintaining computational efficiency and generalizability.

Experimental validation across both simulated and real-world scenarios demonstrates NEAR's superior performance, consistently outperforming baseline methods in response recovery, angular resolution, and direction-of-arrival estimation. The results showcase the method's robustness across different sampling configurations, noise levels, and number of targets, with significant improvements observed particularly in complex multi-target environments. Comprehensive evaluations using commercial MIMO radar platforms further validate the framework's practical applicability and potential for enhancing radar sensing technologies.

**Claims And Evidence:**

**Claim: Extensive simulations and real-world experiments using radar platforms demonstrate NEAR’s effectiveness**

Assessment:  Lack of comparison with data-driven algorithms and lack of computational efficiency comparison experiments make it impossible to determine the method's effectiveness.

**Essential References Not Discussed:**

N/A

**Experimental Designs Or Analyses:**

**Issue 1**: Lack of comparison with data-driven algorithms.

**Issue 2**: Lack of computational efficiency comparison experiments, making it impossible to determine the method's usability.

**Methods And Evaluation Criteria:**

Yes, using simulations and real-word experiments to evaluate method's performance is reasonable.

**Other Comments Or Suggestions:**

N/A

**Other Strengths And Weaknesses:**

N/A

**Questions For Authors:**

1. Please provide detailed experimental comparisons or a comprehensive analysis of your method's advantages over data-driven algorithms in terms of accuracy and computational efficiency.

2. Please provide experimental comparisons or a detailed analysis of the computational efficiency differences between your method and EMaC.

**Relation To Broader Scientific Literature:**

The paper applies cutting-edge AI technology INR to traditional problems, and combines domain-specific prior knowledge as constraints, successfully solving domain problems. This problem-solving approach may potentially be extended to many other fields.

**Theoretical Claims:**

The constraints used in the paper are common constraints in its application domain, and can effectively constrain INR.

---

> ### Author Rebuttal · Authors · 2025-04-01
>
> We sincerely thank the reviewer xAKz for the constructive comments and suggestions. We provide additional experimental comparisons below to address your concerns:
>
> **1. Experimental comparison between our approach and data-driven method (NeRF$^2$).**
>
> We add a state-of-the-art data-driven baseline called NeRF$^2$[1], which represents scenes as neural radiance fields by optimizing an underlying continuous volumetric scene function using a set of input electromagnetic signal measurements. We compare NeRF$^2$ and our method (NEAR) in terms of angular resolution, target localization accuracy (same as we did in Section 5.2) and average running time using the real-world collected data. We adopt hyperparameters recommended in [1], and run all experiments on a laptop with CPU AMD Ryzen 9 5900 HS with Radeon Graphics and GPU NVIDIA GeForce RTX 3050 Ti Laptop.
> | Method  | Angular resolution for 2m/3m/4.5m | Localization error for 1/2/3/4 target(s) | Average running time |
> |:-------:|:--------------------------------:|:--------------------------------------:|:--------------------:|
> | NEAR    | 5.7248$^\circ$/6.6769$^\circ$/6.9941$^\circ$ | 0.0744m/0.0770m/0.0762m/0.0718m | 550.83s |
> | NeRF$^2$ | 8.5783$^\circ$/8.5783$^\circ$/8.8948$^\circ$ | 0.4902m/0.5096m/0.4346m/0.3898m | 1278.31s |
>
> From the table, we can see that NEAR is able to resolve smaller angle separation at different distances (SNR) and achieve a much smaller target localization error, compared to NeRF$^2$, highlighting the effectiveness and efficiency of our method. This improvement is primarily attributed to judicious use of signal processing ideas in designing the regularizer, which fully exploits the underlying harmonic structure in planar wave propagation from far-field targets.
>
> **2. Experimental comparison of computational efficiency between our approach and EMaC.**
>
> For the average running time, EMaC takes **1226.15s** while our approach only takes **550.83s**, indicating the potential of real-time implantation with future algorithmic and computing hardware improvements.
>
> **References:**
>
> [1] Zhao, X., An, Z., Pan, Q., \& Yang, L. (2023, October). Nerf2: Neural radio-frequency radiance fields. In Proceedings of the 29th Annual International Conference on Mobile Computing and Networking (pp. 1-15).

---

### Official Review · Reviewer_MVwc · 2025-03-14

**Overall Recommendation:** 3

**Summary:**

The authors utilize a new INR-based framework to achieve angular super resolution in multi-antennae radar systems. The authors further propose a physics-informed regularizer and provide theoretical insights into what functions can be represented by INRs under certain, in previous literature established, constraints. The authors provide extensive synthetic and real-world results as well as an ablation study, showing the superiority of the proposed approach over other solutions as well as the positive performance impact of individual components.

**Claims And Evidence:**

The performance claims made in this work are sufficiently supported by the experiments conducted. I have not extensively validated the proofs for claims about INRs representational power.

**Essential References Not Discussed:**

I am not familiar enough with the literature in this domain.

**Experimental Designs Or Analyses:**

The ablation study is appropriately designed.

**Methods And Evaluation Criteria:**

I am not familiar with the application domain, as such I cannot speak on whether the evaluation criteria are appropriate. The ablation study is sufficient evidence to at least support baseline claims about the framework's efficacy and necessity of individual components (i.e. the additional regularizer).

**Other Comments Or Suggestions:**

page 3, line 111, right: Typo in "reconstruc" - missing "t"

**Other Strengths And Weaknesses:**

The experiments in this paper seem well-selected and demonstrate the strong performance of the proposed method. On first glance the proofs also seem detailed and, given their veracity, make interesting contributions to the literature that elevates the overall significance of the paper substantially.

**Questions For Authors:**

No questions.

**Relation To Broader Scientific Literature:**

This work extends the literature in two directions, to the best of my knowledge. It introduces two new methods, NEAR broadly and the physics-informed regularizer specifically. It also extends the literature on INRs representational capabilities.

**Theoretical Claims:**

I did not check the correctness of proofs provided in the paper.

---

> ### Author Rebuttal · Authors · 2025-04-01
>
> We sincerely thank the reviewer MVwc for the time and effort in reviewing our paper. We appreciate your positive comments on our work and have fixed the typo you pointed out. If there are any additional areas where you believe we could further improve our manuscript, we would greatly appreciate your insights. Please let us know if we can address any specific concern that could help justify a higher score. Thank you again for your consideration and support.

---

### Official Review · Reviewer_5fdJ · 2025-03-19

**Overall Recommendation:** 4

**Summary:**

Problem Statement:
The paper tackles the challenge of achieving angular super-resolution in multi-antenna radar systems using only sparse measurements. In radar systems, hardware constraints (i.e. having only a few physical antennas) and noise limit the achievable angular resolution. Traditional supervised methods often require large, high-quality training datasets and may not generalize well to new environments.

Proposed Solution (NEAR):
The authors propose NEAR—a framework based on untrained implicit neural representations (INRs) that predicts complex-valued antenna responses at unseen locations from sparse measurements. The method is physics-informed, as it leverages the latent harmonic structure inherent in electromagnetic wave propagation, and it incorporates a novel latent geometry–aware regularizer.

Main Claims:

- High-Resolution Prediction: NEAR can predict full virtual array responses (both amplitude and phase) with high accuracy despite limited measurements.
- Generalization & Interpretability: By integrating physical laws (such as planar wave propagation and harmonic structure), the model generalizes well to unseen environments and remains physically interpretable.
- Superior Angular Super-Resolution: NEAR improves upon conventional methods in resolving closely spaced targets (i.e., achieving super-resolution) while keeping hardware costs low.
- Theoretical Foundations: The paper provides new theoretical insights into the expressive power of INR architectures when equipped with appropriate positional encodings and shows how these relate to the Fourier harmonic representation of array responses.

**Claims And Evidence:**

I found the claims to be well supported.

**Essential References Not Discussed:**

I think the related work section is detailed and comprehensive.

**Experimental Designs Or Analyses:**

Simulation Studies:

- Setup: The simulations use virtual antenna array responses under various SNR conditions and different sparse sampling patterns (e.g., 6×6, 8×8, 10×10 grids).
- Baselines: NEAR is compared against Enhanced Matrix Completion (EMaC), a SIREN-based approach, and a variant of NEAR without the physics-informed regularizer.
- Results:
NEAR consistently achieves lower NRMSE across various SNR levels and sampling patterns.
In angular resolution tests, NEAR maintains high resolution probability even at small angle separations.
For multi-target DOA estimation, NEAR shows significantly lower estimation errors than baselines, demonstrating its robustness in complex scenarios.

Real-World Experiments:

- Setup: A commercial MIMO radar platform (IMAGEVK-74) with a 20×20 virtual array is used. Sparse subsets of the full array response are treated as input.
- Evaluation:
- Angular Resolution: Experiments with two corner reflectors measure the minimum resolvable angular separation at various distances. NEAR achieves performance close to the full array benchmark.
- Target Localization: The system is tested with multiple reflectors. NEAR outperforms both the full array baseline and EMaC in terms of localization error, attributed to its denoising capability.

- Ablation Study:

Removing the physics-informed regularizer leads to significantly worse performance, underscoring the importance of incorporating physical constraints into the model.

**Methods And Evaluation Criteria:**

The methods are described well and the authors do experimental studies on simulated as well as real world datasets with impressive results.

- Implicit Neural Representation (INR):

NEAR employs an untrained INR that maps 2D spatial coordinates to complex-valued radar responses.
The architecture follows a typical INR design: a multilayer perceptron (MLP) with positional encoding (similar to NeRF) is used to represent the continuous response field over a virtual antenna array.

- Physics-Informed Regularization:

A key innovation is the introduction of an implicit regularizer that exploits the harmonic and low-rank structure of the radar array response.
The authors show that when the response field is expressed in a domain where the underlying physics (planar wave propagation) holds, its corresponding block Hankel matrix exhibits low rank.
The regularizer is integrated into the loss function to enforce consistency with these known physical properties.

- Loss Function and Training:

The overall loss is composed of a data fitting term (quantifying the difference between the predicted response and the available sparse, noisy measurements) and a regularization term (enforcing the harmonic/low-rank structure).
Importantly, NEAR is trained without extensive offline training data—it uses only the sparse measurements obtained during normal operation.

- Theoretical Analysis:

The paper provides rigorous theoretical results that characterize the class of functions representable by the INR architecture, linking it to Fourier series.
This analysis not only justifies the choice of positional encoding and network architecture but also explains how the harmonic structure of radar signals can be effectively captured.

**Other Comments Or Suggestions:**

N/A

**Other Strengths And Weaknesses:**

N/A

**Questions For Authors:**

I don't have major concerns.

One question is: In the regularized loss function, I don't see the point of having $m_1, m_2$ as variables. For each $\theta$, you can compute the prediction $\widehat{Y}$ and use the pseudoinverse projection to compute $m_1, m_2$ in closed form. Why would you run an optimizer on $m_1, m_2$ as well?

**Relation To Broader Scientific Literature:**

The paper seems like a good contribution. The proposed method highlights the challenges associated with employing INR models to RADAR, and designs regularizers that respect the harmonics present in RADAR. The loss function designed is straightforward, but I think the authors do a good job evaluating their method and comparing it to baselines on simulated and real-world data. The significance of the Theorems is doubtful -- they seem unsurprising.

**Theoretical Claims:**

I did not check the correctness of the proof. The theorem statements are sensible, although I'm not sure why Theorem 4.5 is needed -- if the Hankel matrices are low rank and the first K columns are independent, then doesn't the statement of the theorem automatically follow from the definition of rank? (this is a minor concern, I don't have major issues with it being included)

---

> ### Author Rebuttal · Authors · 2025-04-01
>
> We sincerely thank the reviewer 5fdJ for the time and effort in reviewing our paper. We greatly appreciate the positive feedback. We hope the following responses can resolve your questions and concerns.
>
> **1. I'm not sure why Theorem 4.5 is needed --- if the Hankel matrices are low rank and the first $K$ columns are independent, then doesn't the statement of the theorem automatically follow from the definition of rank? (this is a minor concern, I don't have major issues with it being included)**
>
> There are two values which Theorem 4.5 add. Firstly, the low-rank property of $\mathcal{H}\_{N\_1,N\_2}(\boldsymbol{Y})$ depends on the choice of $N\_1$ and $N\_2$ according to the theoretical results from [1]. However, we cannot directly use the sufficient condition from [1] to claim that $\mathcal{H}\_{M\_1,M\_2-K}(\boldsymbol{Y})$ is rank-$K$ since in this case $N\_1=M\_1\notin[K,M\_1-K+1]$. Secondly, Theorem 4.5 reveals an important relation between submatrices of this low-rank matrix. It shows that all small Hankel matrices $\mathcal{H}\_{M\_2-K}(\mathbf{y}\_{m})\text{ for }1\leq m\leq M\_1$ are not only rank-$K$ with their first $K$ columns serving as a basis, but also share the *same linear coefficients $\boldsymbol{m}\_1$*. More details can be found in Appendix B. This is a significant result, since it helps us to re-use the parameters $\boldsymbol{m}\_1$ for representing subsequent new entries at unseen locations. Such theoretical results are non-trivial and no existing results can be found in the literature according to our best knowledge.
>
> **2. The significance of the Theorems is doubtful --- they seem unsurprising.**
>
> We mainly have two theoretical results, one is about expressive power of INRs, the other is about our physics-informed regularizer. We obtain a tighter characterization on the expressability of INR by refining existing analysis and derive the exact set of integer harmonics which describe the expressive power of INRs. One the other hand, our regularizer utilizes physical model (planar wave propagation) and mathematical properties of harmonics structured matrices, which establish a connection between low rank and linear predictability. This results in a regularizer which is both computationally efficient and significantly improves overall performance, as shown by our experimental studies.
>
> **3. One question is: In the regularized loss function, I don't see the point of having $\boldsymbol{m}\_1,\boldsymbol{m}\_2$ as variables. For each $\boldsymbol{\theta}$, you can compute the prediction $\hat{\boldsymbol{Y}}$ and use the pseudoinverse projection to compute $\boldsymbol{m}\_1,\boldsymbol{m}\_2$ in closed form. Why would you run an optimizer on as well?**
>
> Thank you for the good question. Please note that $\boldsymbol{\theta}$ and $\boldsymbol{\phi}$ are angles of arrival from multiple point scatterers, and do not represent incident angles. The unique global optimal $\boldsymbol{m}\_1^o$ and $\boldsymbol{m}\_2^o$ depend on target angles $\boldsymbol{\theta}$ and $\boldsymbol{\phi}$, which are part of the sensing task and not known beforehand. In fact, the goal is to first complete a virtual array (by predicting array response at unseen locations) and then use the physical and predicted measurements to obtain a more accurate estimate of the angles $\boldsymbol{\theta}$ and $\boldsymbol{\phi}$ (and not the other way). If we were to first estimate the angles  $\boldsymbol{\theta}$ and $\boldsymbol{\phi}$ from a *limited number of sparse* physical antennas (without completing the virtual array), the estimation error would be much larger. Therefore, in order to predict the virtual array response, we use the combined power of INR and the latent variables $\boldsymbol{m}\_1$ and $\boldsymbol{m}\_2$, which exploits low-rank relationship between entries of the Block Hankel matrix and guides the INR to approach the ground truth array response. However, if $\boldsymbol{m}\_1$ and $\boldsymbol{m}\_2$ are computed using the pseudoinverse projection after predicting $\hat{\boldsymbol{Y}}$ purely using INR, then the whole algorithm runs the risk of overfitting the  observed data, especially given the scarce number of spatial measurements.
>
> **References:**
>
> [1] Hua, Y. (1992). Estimating two-dimensional frequencies by matrix enhancement and matrix pencil. IEEE Transactions on Signal Processing, 40(9), 2267-2280.

---

### Decision · Program_Chairs · 2025-05-01

**Decision:**

Accept (poster)

**Comment:**

The paper received consistently positive feedback following the authors’ rebuttal. Reviewers highlighted the paper's theoretical foundation (5fdJ, moMV, MVwc), the quality of the work (MVwc, 5fdJ) with promising results. While there were initial concerns about the lack of comparisons regarding data-driven algorithms and computational efficiency, these were effectively addressed in the rebuttal, leading to stronger overall reviews.

After carefully reviewing the paper, the reviews, and the rebuttal, The AC concurs with the reviewers' positive consensus and recommends acceptance of this paper.

For the camera-ready version, the authors should integrate all discussions from the rebuttal into the main paper and supplementary materials. Specifically, the authors should implement the following changes:
1. Experimental comparison between our approach and data-driven method (NeRF2)
2. Experimental comparison of computational efficiency between our approach and EMaC.